# xLSTM Scaling Laws: Competitive Performance with Linear Time-Complexity

**Maximilian Beck** [1,2]    **Kajetan Schweighofer** [1]
**Sebastian Böck** [2]    **Sebastian Lehner** [1]    **Sepp Hochreiter** [1,2]

[1] ELLIS Unit Linz, Institute for Machine Learning, JKU Linz, Austria
[2] NXAI GmbH, Linz, Austria
{beck,schweighofer,slehner}@ml.jku.at

## Abstract

Scaling laws play a central role in the success of Large Language Models (LLMs), enabling the prediction of model performance relative to compute budgets prior to training. While Transformers have been the dominant architecture, recent alternatives such as xLSTM offer linear complexity with respect to context length while remaining competitive in the billion-parameter regime. We conduct a comparative investigation on the scaling behavior of Transformers and xLSTM along the following lines, providing insights to guide future model design and deployment. First, we study the scaling behavior for xLSTM in compute-optimal and over-training regimes using both IsoFLOP and parametric fit approaches on a wide range of model sizes (80M-7B) and number of training tokens (2B-2T). [1] Second, we examine the dependence of optimal model sizes on context length, a pivotal aspect that was largely ignored in previous work. Finally, we analyze inference-time scaling characteristics. Our findings reveal that in typical LLM training and inference scenarios, xLSTM scales favorably compared to Transformers. Notably, xLSTM models consistently Pareto-dominate Transformer models, delivering lower cross-entropy loss for the same compute budget.

## 1 Introduction

Scaling up models sizes and training data sets enables the recently observed rapidly advancing capabilities of Large Language Models (LLMs). As a result the computational expenses associated to training and inference of state-of-the-art LLMs results are dramatically growing. The goal of predicting the achievable performance with a specified architecture and computational resources resulted in the recent exploration in LLM scaling laws, i.e. the quantitative relationships between LLM performance metrics and the corresponding computational resources. The works of Kaplan et al. (2020); Hoffmann et al. (2022) showed that these scaling laws take the form of power laws which hold over several orders of magnitude in terms of model sizes and the number of pre-training tokens. These insights provided practical guidance in the design of recent frontier models (Achiam et al., 2023; Grattafiori et al., 2024; DeepSeek-AI, 2024a).

Recent works (Sardana et al., 2024; Gadre et al., 2024) rightfully argue that these scaling laws are nevertheless limited by their neglect of inference costs. Consequently, these works focus on performance investigations on models that are trained in the so-called over-training regime, i.e. on more tokens than would be optimal in terms of pre-taining compute. Importantly, these works and subsequent ones focus on Transformer architectures (Vaswani et al., 2017). In these architectures, the attention mechanism inflicts computational costs during training and inference that are *quadratic* in terms of context length. Besides the associated economic and ecological costs, this quadratic scaling is prohibitive for a large range of application areas in which models are deployed on devices with limitations on available memory, energy, or allowable TFTT. Even on GPUs that are dedicated to LLMs this scaling property of Transformers represents a limitation in task that require

---

[1]The code and data to reproduce our analyzes and figures is available at:
https://github.com/NX-AI/xlstm_scaling_laws

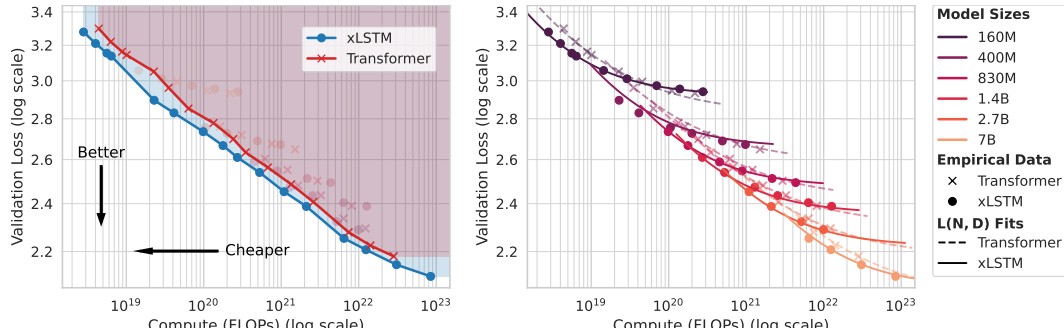

Figure 1: xLSTM scaling laws: Validation loss over training compute. **Left:** xLSTM is pareto-dominant over dense multi-head Transformers in terms of loss. For a fixed FLOP budget, xLSTM models are better. For a fixed validation loss, xLSTM models require less FLOPs. **Right:** Parametric fit of the loss surface $L(N, D)$ as a function of model size $N$ and dataset size $D$.

very long contexts, like reasoning (Muennighoff et al., 2025). Consequently, the development of LLM architectures that mitigate the attention mechanism is an active area of research (Gu & Dao, 2024; Beck et al., 2024; Lieber et al., 2024). While these architectures were demonstrated to be scalable into the billion-parameter regime (Zuo et al., 2024; Beck et al., 2025b), there is so far no systematic comparison between linear complexity LLM architectures, i.e. LLMs that scale linearly in computational costs with respect to context lengths, and transformer-based LLMs with quadratic complexity.

This work presents a systematic comparison of the scaling laws of performance-optimized xLSTM architectures (Beck et al., 2025b;a) and dense multi-head self-attention Transformer architectures (Touvron et al., 2023). Our investigations of xLSTM and Transformer models are guided by the following research questions:

- *Training*: Which architecture can be trained more efficiently in terms of computational resources and how do they scale in the practically relevant overtaining regime?

- *Context length*: How does the striking difference between xLSTM and Transformers—linear versus quadratic context length dependency—impact scaling laws and the resulting pre-training and inference performances?

- *Inference*: How does the inference speed in terms of time to first token (prefill) and step time (generation) scale for xLSTM and Transformer under different context lengths and model sizes?

Our investigation shows, that **xLSTM models Pareto-dominate Transformer models in the compute–loss trade-off** (Fig. 1), enabling models that are both better and cheaper. We find that, for a given training compute budget, compute-optimal xLSTM models are larger (Fig. 4), i.e. have more parameters, than compute-optimal Transformer models. During inference, xLSTMs are faster than same-sized Transformers (Fig. 6), and their performance advantage grows with context length due to Transformers' quadratic time complexity.

## 2 PRELIMINARIES

We begin with a background on scaling laws and a definition of the training regimes considered in this work (Sec. 2.1). We next present approaches for scaling law fitting used in this study (Sec. 2.2).

### 2.1 BACKGROUND ON SCALING LAWS

Scaling laws for large language models predict the cross-entropy loss $L$ as a function of the compute $C$ used for model training in FLOPs. The compute $C$ for training increases with larger model size measured in number of model parameters $N$ and larger dataset size in number of training tokens $D$. Hence, we assume $C$ is a function of $N$ and $D$. Depending on how the total compute budget is distributed between increasing the model size and enlarging the dataset, training is typically characterized as either being in a *compute-optimal* or in an *over-training* regime.

**Compute-optimal training.** Hoffmann et al. (2022) establish the notion of compute-optimal training, which refers to the optimal choice of $N$ and $D$ for a given compute budget $H$ according to the constrained optimization problem:

$$N^*(H), D^*(H) = \underset{N,D \text{ s.t. } C(N,D)=H}{\text{argmin}} L(N, D). \tag{1}$$

The optimal $N^*$ and $D^*$ can be obtained by sweeping over $N, D$ for each compute budget. Hoffmann et al. (2022) find that for increasing computation budgets, $N^*$ and $D^*$ scale roughly proportionally. Assuming this proportionality, there exists a compute-optimal token per parameter ratio $M^* = D^*/N^*$ for a fixed model class and training distribution.

**Over-training.** The compute-optimal allocation $D^*$, $N^*$ only accounts for compute costs during training. However, during inference larger models incur a higher inference compute cost. Taking this into account, Sardana et al. (2024) argue that, once inference costs are considered, it can be preferable to train smaller models on larger datasets. The resulting values for $D$ and $N$, with a higher than compute-optimal token per parameter ratios $M > M^*$ is generally referred to as *over-training* regime (Gadre et al., 2024).

**Calculating compute costs.** Previous works on transformer scaling laws commonly approximate compute costs with $C(N, D) = 6ND$ FLOPs (Kaplan et al., 2020; Hoffmann et al., 2022; Gadre et al., 2024; Sardana et al., 2024). This approximation ignores the FLOPs associated to the attention mechanism and covers only the feed-forward network contributions. Recently, several works (DeepSeek-AI, 2024a; Busbridge et al., 2025; Li et al., 2025) pointed out that this approximation is not justified for sufficiently large context lengths and models. For the purpose of this work, this approximation is even less suitable since it neglects entirely the difference between linear and quadratic time-complexity models. Hence, we adopt a more precise calculation of $C(N, D)$ as provided in Appendix B.3 that accurately captures the differences in computational complexity between model classes.

## 2.2 FITTING SCALING LAWS

Scaling laws are obtained by fitting the dependence of the model's training or validation loss on the model size and the number of training tokens with power laws. Two commonly used procedures for extracting parametric scaling laws for the loss L, depending on $N$ and/or $D$ are the *parametric fit approach* and the *IsoFLOP approach*, which are introduced in Hoffmann et al. (2022) as the third and second approach, respectively.

**Parametric fit approach.** Assuming that the loss $L$ follows a power law in model parameters $N$ and training tokens $D$, the parametric fit approach estimates the observed cross-entropy loss as:

$$\hat{L}(N, D) = E + (A \, N^{-\alpha} + B \, D^{-\beta})^\gamma, \tag{2}$$

where $E, A, B, \alpha, \beta$, and $\gamma$ are task-specific positive parameters. The constant term $E$ accounts for an irreducible loss component, while the second term captures the model-specific predictive performance. While Hoffmann et al. (2022) set $\gamma = 1$, we follow the practice from Busbridge et al. (2025) and treat $\gamma$ as fit parameter.

A robust estimation of the scaling parameters for (2) requires data from diverse training strategies, including non-compute optimal token-to-parameter ratios. Therefore, Hoffmann et al. (2022) include data from two training strategies: (i) The number of training tokens is varied for a fixed set of models. (ii) Model size and training tokens are both varied subject to a total compute constraint.

**IsoFLOP approach.** For the IsoFLOP approach a set of compute budgets $H$ is defined and for each budget the values of $N$ and $D$ are varied such that the constraint $C(N, D) = H$ is fulfilled. Following Hoffmann et al. (2022), a second-order polynomial is fitted to each of the resulting IsoFLOP profiles. The minimum of each fit corresponds to the loss-optimal number of model parameters $N^*(H)$ and training tokens $D^*(H)$ for the given compute budget $H$. In order to predict these quantities, we use individual power laws of the forms

$$\hat{N}^*(H) = A' \cdot H^a \qquad \text{and} \qquad \hat{D}^*(H) = B' \cdot H^b, \tag{3}$$

where we fit the *exponents* $a, b$ and *coefficients* $A', B'$ from the data.

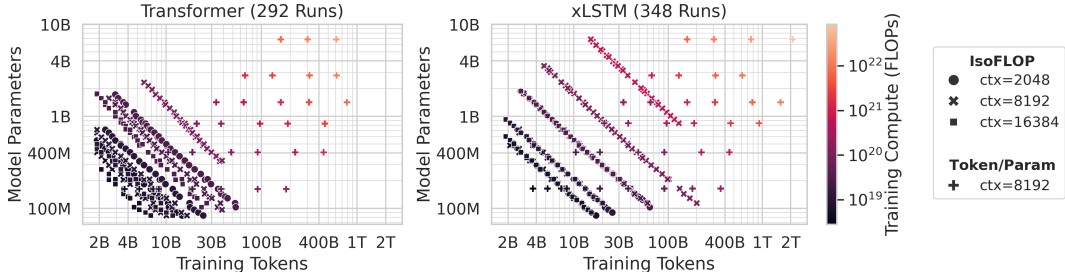

Figure 2: Dataset of training runs for our scaling law study. The dataset contains training runs for the xLSTM and the Transformer architecture, with two configurations each: *IsoFLOP* and *Token/Param*.

## 3 TRAINING SCALING BEHAVIOR

In this section, we conduct a comparative study of the scaling behavior of xLSTM and Transformer models along multiple axes. First, we explore the pareto frontier of performance in terms of loss and training compute in Section 3.2. Second, we study the scaling in the over-training regime with large token to parameter ratios in Section 3.3. Finally, we determine the compute-optimal model and dataset sizes in Section 3.4 and their dependence on the context length in Section 3.5. We begin with the introduction of our experimental setup in Section 3.1.

### 3.1 EXPERIMENTAL SETUP

To systematically study scaling behavior, we collect a large dataset of training runs across two model classes (Transformer and xLSTM) and multiple training configurations. The following describes the architectures, training recipe, and dataset of training runs used in our scaling law study.

**Model architectures: Transformer and xLSTM.** Following previous scaling law studies (Porian et al., 2024; Gadre et al., 2024), we use the dense multi-head attention decoder-only Llama-2 architecture (Touvron et al., 2023) for our Transformer models. For the xLSTM models, we consider the architecture of the recently proposed xLSTM 7B model (Beck et al., 2025b). The xLSTM-7B architecture is built entirely on mLSTM cells with parallel training mode applied within the model's embedding dimension. Similar to the Transformer, it alternates mLSTM layers with position-wise feedforward MLP layers. The crucial distinction between the two architectures lies in the sequence-mixing mechanism: self-attention with quadratic time-complexity in Transformer versus recurrent mLSTM dynamics with linear time-complexity in xLSTM.

**Training recipe and data.** For both model classes we use the same training recipe derived from the xLSTM 7B training recipe (Beck et al., 2025b). The recipe uses the AdamW optimizer ($\beta_1 = 0.99$, $\beta_2 = 0.95$, $\epsilon = 10^{-8}$), weight decay $0.1$ and gradient clipping norm $0.5$. The learning rate scheduler has three stages, linear warm-up, cosine decay to $10\%$ of the peak learning rate, and linear cool-down. For varying compute budgets, we scale the steps in the second stage while the first and third remain fixed. Further details are given in Appendix A.1. The overall number of training steps is determined by the FLOP budget or token-to-parameter ratio of the specific experiment. As training dataset, we use DCLM-BASELINE, a collection of high-quality filtered web documents (Li et al., 2024), tokenized with the GPT-NeoX tokenizer (Black et al., 2022) into sequences of length 8192, unless specified otherwise. We use `grain`[2] to prepare batches with sequence packing, particularly first-fit packing, which avoids splitting, but adds padding tokens.

**Dataset of training runs.** Using the above defined architecture and training recipe, we produce a large dataset of training runs for our scaling law study totaling 640 individual runs (292 for Llama, 348 for xLSTM). The dataset contains model sizes ranging from 80M to 7B parameters trained with compute budgets ranging from $2.8 \times 10^{18}$ to $8.5 \times 10^{22}$ FLOPs on 2B to 2T tokens. This amounts to a total compute budget spent for this dataset of $3.2 \times 10^{23}$ FLOPs. Our dataset is divided in into runs from two different training configurations: *IsoFLOP* and *Token/Param*. For the IsoFLOP configuration, we vary model parameters and training tokens subject to fixed compute budgets for

---

[2]https://google-grain.readthedocs.io ( FirstFitPackIterDataset)

Figure 3: Power law fits to loss over training compute with increasing token-to-parameter (Token/Param) ratios $M$. We fit power laws of the form in $\hat{L}(C) = \lambda \cdot C^{-\eta}$ and observe that— similar to Transformer—the exponents $\eta$ of xLSTM remain constant even for large $M$, indicated by the parallel lines in the log-log plot.

three different context lengths. In the Token/Param configuration, we vary the number of training tokens for a set of fixed model sizes. We show our dataset as $\{N, D, C\}$ points in Figure 2. xLSTM's linear scaling preserves training tokens with longer contexts (overlapping IsoFLOP points), whereas Transformer's quadratic scaling reduces them.

## 3.2 LOSS VS. COMPUTE: xLSTM IS PARETO-DOMINANT

We begin our study with the question: Given a fixed training compute budget, which model architecture performs better (in terms of cross-entropy loss)? To answer this question, we define a grid of model and dataset sizes with pre-defined token-to-parameter ratios of $[22, 44, 110, 220, 550, 1100, 2200]$ and train Transformer and xLSTM models for each point in the grid. This forms the *Token/Param* subset in our dataset of training runs (see Sec. 3.1). We then use our FLOP calculations in Appendix B.3 and plot validation loss over FLOPs in a log-log plot in Figure 1.

**Pareto-frontier.** In Figure 1 (left), we visualize the Pareto frontier by connecting the data points for xLSTM and Transformer. We find that xLSTM is strictly dominant over Transformers across the almost five orders of magnitude of compute encompassed by our data. In other words, for a fixed FLOP budget, xLSTM models are better and for a fixed validation loss, they require less FLOPs.

**Parametric loss surface fit.** In Figure 1 (right), we fit a parametric loss surface $\hat{L}(N, D)$ to our Token/Param data. We find that our fit of the loss surface provides a reliable description of performance of Transformer and xLSTM models for a given size even far in the over-training regime, i.e. far right to the pareto front. Following the practice of Busbridge et al. (2025), we find that including the parameter $\gamma$ in the model of $\hat{L}(N, D)$ improves the fit quality (see Fig. 8 in the Appendix). We provide additional details on our parametric fits in Appendix A.2.

## 3.3 xLSTM IN THE OVERTRAINING REGIME: CONSISTENT POWER LAW EXPONENTS

Our parametric $\hat{L}(N, D)$ fit predicts, that model quality in terms of loss improves when $N$ or $D$ is increased. Hoffmann et al. (2022) have found that for Transformers, the optimal token-to-parameter ratio $M^* = D^*/N^*$ that yields the minimal loss under a compute constraint is approximately 22. However, training runs with this ratio yield rather large models that are expensive and slow during inference (Sardana et al., 2024). Consequently, it is common practice to train smaller models in an overtraining regime, i.e., with token-to-parameter ratios far exceeding the compute-optimal $M^*$. It is thus of practical importance to demonstrate that the loss of new model architectures continues to improve with increasing amounts of data.

**Power-law exponents in over-training.** Gadre et al. (2024) have found that Transformers scale reliably in this over-training regime, indicated by constant exponents $\eta$, when fitting a power law of the form $\hat{L}(C) = \lambda \cdot C^{-\eta}$ for different fixed token-per-parameter ratios $M$. Therefore, we perform a similar analysis and fit power laws $\hat{L}(C)$ to our Token/Param training runs. In Figure 3 and Tab. 3 we find that — similar to Transformer — the exponents $\eta$ of xLSTM remain constant even for large $M$, indicated by the parallel lines in the log-log plot. This observation is relevant because it implies that small, inference-optimized xLSTM models can be trained on large datasets while still achieving consistent improvements in loss.

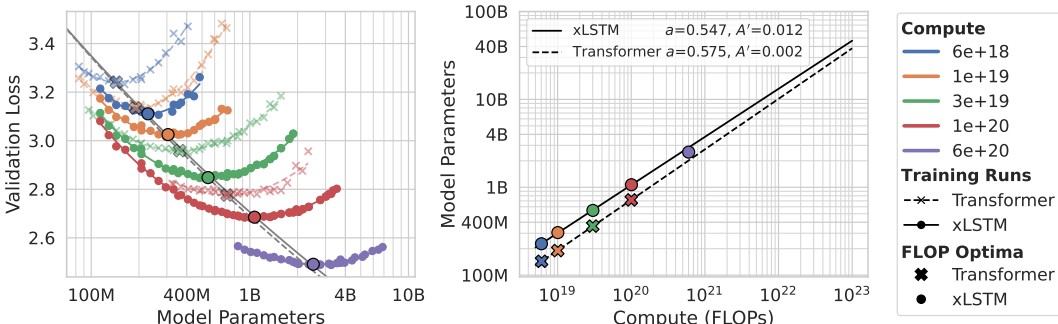

Figure 4: Varying model size and tokens with a fixed compute budget (IsoFLOP). **Left:** IsoFLOP profiles for varying number of model parameters with a marker at the minimum $N^*$ of the fitted polynomial. **Right:** Power-law fit $N^*(H) = A' \cdot H^a$ for the compute optimal number of model parameters. Our setup reproduces the power-law exponent $a$ for Transformers established in Porian et al. (2024). The compute-optimal model size of xLSTMs is larger than for Transformers.

## 3.4 COMPUTE-OPTIMAL xLSTM MODELS ARE LARGER

In this section, we aim to determine the compute-optimal model size $N^*$ and dataset size $D^*$ for the xLSTM and Transformer models. However, so far, we have performed our scaling analyses on training configurations with preset model sizes and a set of token-per-parameter ratios $M$, which do not allow us to determine $N^*$ and $D^*$ directly. Therefore, for this analysis, we use the *IsoFLOP* training configuration, where we vary the number of model parameters and training tokens subject to a set of fixed compute budgets $H$. For each compute budget, we plot the loss over the model parameters $N$ and number of training tokens $D$ and fit second-order polynomials to determine the optimal $N^*(H)$ and $D^*(H)$ for each compute budget $H$. Using these optima, we then fit power laws as described in Section 2.2 to obtain the functional forms for $\hat{N}^*(H)$ and $\hat{D}^*(H)$ (see Eq. (3)).

**Compute-optimal model size.** In Figure 4 (left) we show the IsoFLOP profiles for variable model size and (right) the corresponding power-law fits for the optimal model size for xLSTM and Transformer. Our results show that for a given compute budget, xLSTM consistently attains a lower validation loss than Transformer, which is in line with the findings in Section 3.2. Moreover, we find that for a given compute budget, the corresponding compute-optimal xLSTM models have more parameters than the corresponding Transformer models; see Figure 4 (left and right). Note that our power-law exponent $a$ for the Transformer matches the one found by Porian et al. (2024); see App. A.4 for details.

**Compute-optimal dataset size.** Analogous results are shown in Figure 9 in the appendix for the number of training tokens of compute-optimal models. We find that compute-optimal xLSTM and Transformer models are trained on a similar number of training tokens $\hat{D}^*(H)$. In Appendix E, we show the estimated optimal training FLOPs and training tokens for various model sizes.

**Universality of the relation between compute-optimal performance and model size.** The compute-optimal models in Figure 4 (left) fall close to a single shared line for the Transformer and xLSTM models. This suggests that for compute-optimal models, there is a universal relationship between performance and model size for xLSTM and Transformer models. From this perspective, the fact that compute-optimal xLSTM models are larger for a given compute budget can be regarded as a heuristic explanation for the superior performance of xLSTM. The reason why xLSTMs can be larger is the reduced computational complexity of their recurrent sequence-mixing operation compared to the self-attention operation in Transformers. As this main operation is cheaper, more compute can be allocated to the rest of the model, e.g. increased number of layers or embedding dimension.

## 3.5 COMPUTE-OPTIMAL xLSTM MODEL SIZE REMAINS STABLE ACROSS CONTEXT LENGTHS

The main difference between the model architectures in this study is their scaling in FLOPs with context length: Transformers scale quadratically, due to the self-attention, while xLSTMs scale linearly. This implies that, in Transformers, an increasing fraction of compute is devoted to attention as sequence length grows, whereas in xLSTMs the recurrent updates consume only a modest portion

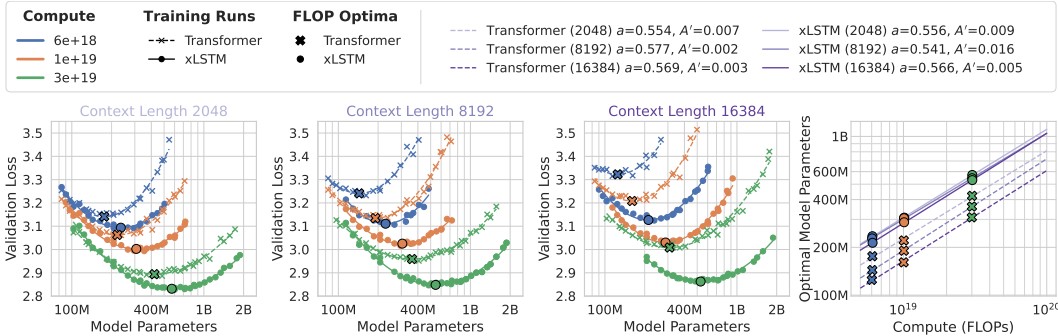

Figure 5: Left: IsoFLOP curves as a function of model parameters at 3 different context lengths. Right: Plot of the power-law fits for the compute optimal number of parameters dependent on the compute budget $N^*(H)$. Colors indicate compute budget and marker types indicate the model types. The compute optimal model size for Transformers gets smaller for larger context lengths, while the compute optimal model size for xLSTM remains similar across context lengths.

of the total compute. In this section, we investigate, therefore, the impact of the context length on compute-optimal model and dataset sizes. We add experiments with context lengths 2048 and 16384 in the IsoFLOP training configuration and then fit the power-laws to each context length for both models, analogously to Section 3.4. We note that the losses are not directly comparable across different context lengths since we use sequence packing for the construction of our training and validation datasets. Hence, for larger context lengths, longer documents can be packed into a batch, effectively changing the data distribution.

**Context length & compute-optimality.** In Figure 5 we show the IsoFLOP profiles for varying model sizes and three different context lengths and compute budgets, including their power-law fits $\hat{N}^*(H)$ in the rightmost plot. We observe that with increasing context lengths the compute-optimal model size of Transformers drops significantly, while for xLSTM it drops only mildly. These results suggest that for Transformers, a growing fraction of compute is consumed by attention operations as sequence length increases, whereas in xLSTMs most FLOPs remain allocated to depth and hidden dimensions. In Figure 10 in Appendix A.5 we show the corresponding IsoFLOP profiles and power-law fits $\hat{D}^*(H)$ for the optimal number of training tokens. We observe similar trends as for the model size: The compute-optimal number of training tokens decreases markedly with larger context length for Transformer models and for xLSTM it slightly increases.

## 4    INFERENCE SCALING BEHAVIOR

The scaling laws analysis in Section 3 is motivated by the goal of the optimal design of pre-training runs for LLMs. However, these considerations neglect inference efficiency. When deploying LLMs at large scale, inference costs and performance are critical aspects. Hence Pope et al. (2023) investigate the inference efficiency of transformer-based LLMs in terms of three criteria: compute, latency, and throughput. More recently Sardana et al. (2024) provided a scaling law analysis of Transformers that extend the pre-training compute optimality consideration (Eq. (1)) to also account for inference compute. This work presents an even more comprehensive analysis in terms of the attainable latency, i.e., time to first token, and the step time during generation. We complement our empirical findings with a quantitative model of a *lower bound* on time to first token and step time, using the detailed calculation of FLOPs (App. B.3) and MemOps (App. B.4) for both model architectures.

**Inference stages.** Typically, large-scale LLM inference is split into the *prefill* and the *generation* stage (Austin et al., 2025; Pope et al., 2023; DeepSeek-AI, 2024b). In the prefill stage the LLMs process the prompt, compute the logits for the first token to be generated, and store the intermediate internal representations of the prompt, i.e. the KV cache for Transformer models or the mLSTM cell states for xLSTM. In the generation stage a token is sampled according to the logits and then the internal representations of the previous tokens in the context window are updated to account for the new token. The generation procedure is repeated for a certain budget or until the end-of-sequence token is sampled. In the following, we investigate the prefill and generation performances separately.

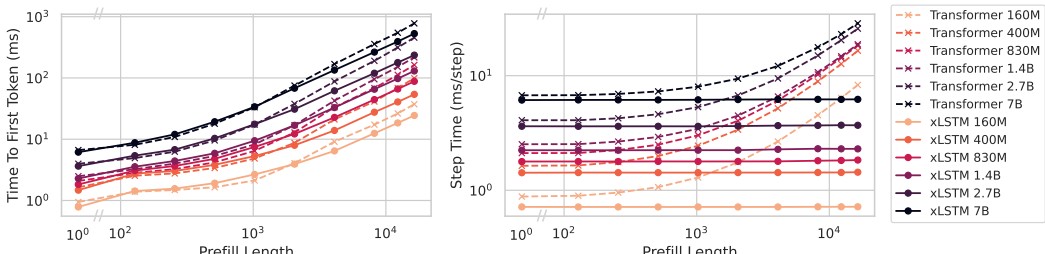

Figure 6: Scaling of TTFT (left) and step time (right) as a function of prefill length (1-16k) for different model sizes, with a batchsize of one.

**Inference runtime metrics.** For the prefill stage, the key performance metric is the time to first token (TTFT). Prefill speed is primarily determined by how well the model can maintain a low TTFT while handling large batch sizes and long input sequences. During the generation stage, the key performance metric is the step time, i.e. how long it takes to obtain the next token given the current (potentially batched) sequence. For Transformers, the quadratic complexity of the attention mechanism with respect to the prefill length (App. B.3.3) implies that TTFT is expected to scale quadratically in terms of the prefill length. In terms of step time we expect linear scaling with respect to the prefill length, as each decoding step involves attention over the entire KV cache. For xLSTMs, in contrast, we expect linear scaling of TTFT and step time that is independent of the prefill length.

### 4.1 EMPIRICAL INFERENCE RUNTIMES

We consider the same model architectures as in the *Token/Param* configuration (see Tab. 19 and 20). We utilize the implementation of xLSTM and Transformers models available through the `transformers` library (Wolf et al., 2020) and optimize runtimes using `torch.compile` and `torch.cuda.graph`. The TTFT is measured as the time needed for generating a single token under a given batch size and prefill length (i.e., the context length). The step time is measured by generating a sequence of 100 tokens, subtracting the TTFT and dividing by the sequence length. We measure the average TTFT and step time over four repetitions after two warm-up iterations.

Figure 6 presents TTFT (left) and step time (right) measurements for both architectures at matched model sizes as a function of prefill length (1-16k). At short prefills, the two model classes exhibit comparable TTFTs, while at longer prefills xLSTMs consistently achieve lower values. For 16k prefill, *xLSTM has 30-50% lower TTFT for the same model size*. This difference reflects the expected scaling: quadratically for Transformers and linearly for xLSTMs. A similar trend is observed for the step time. At small prefills, both architectures perform comparably. As the prefill length increases, the Transformer step time degrades due to the rising cost of attention over longer KV caches. In contrast, xLSTM step time is independent of prefill length, resulting in consistently higher throughput across all evaluated model sizes and prefill lengths. For 16k prefill, *the largest xLSTM has a lower step time than the smallest Transformer* we considered. In summary, when matched in model size, xLSTMs outperform Transformer models on all inference speed metrics considered.

### 4.2 MODELING INFERENCE RUNTIMES

In our analysis, the inference processes are characterized by the associated number of floating point operations $\text{FLOPs}_{\text{algo}}$ and the number of memory operations $\text{Bytes}_{\text{mem,algo}}$ measured in bytes that are read or written. We provide calculations of these two quantities for xLSTM and for Transformers in Appendix B. Importantly, these calculations capture the difference between xLSTM and Transformers in the dependence of $\text{FLOPs}_{\text{algo}}$ and $\text{Bytes}_{\text{mem,algo}}$ on the context length $T$. Based on these calculated quantities, we model the runtimes associated with the floating point and memory operations as:

$$\tau_{\text{FLOPs,algo}} = \frac{\text{FLOPs}_{\text{algo}}}{\alpha_{\text{eff}}} + \epsilon, \qquad \tau_{\text{mem,algo}} = \frac{\text{Bytes}_{\text{mem,algo}}}{\beta_{\text{eff}}} + \epsilon, \qquad (4)$$

where $\alpha_{\text{eff}}$ is the effective rate of FLOPs/s, $\beta_{\text{eff}}$ is the effective rate of Bytes/s, and $\epsilon$ is a constant overhead when running the inference processes on the GPU. Depending on the model type, model size, prefill length, batch size and inference stage (prefill or generate), either $\tau_{\text{FLOPs,algo}}$ or $\tau_{\text{mem,algo}}$ is

the dominant contributor to the runtime. We outline in Appendix C.1 how this is determined based on the roofline model. Using empirical runtime measurements, we then fit one of the two models depending on which one is expected to yield the dominant runtime contribution. Each fit corresponds to a specific model type, size, and inference stage, and is evaluated over varying batch sizes and prefill lengths. As evidenced by the fits to empirical TTFT (App. C.2) and step time measurements (App. C.3), our model provides an accurate description of the observed inference runtimes for both architectures and explains the empirically observed runtimes in Figure 6.

## 5 RELATED WORK

**Modeling scaling behavior with parameters and data.** The empirical scaling behavior of Deep Learning models w.r.t the size of their model parameters and training data has been actively researched (Hestness et al., 2017; Rosenfeld et al., 2020; Henighan et al., 2020; Alabdulmohsin et al., 2022; Caballero et al., 2023). Such scaling laws have been demonstrated across many tasks and data modalities (Tan & Le, 2019; Ghorbani et al., 2022; Zhai et al., 2022; Abnar et al., 2022; Ardalani et al., 2022; Gao et al., 2023) However, beginning with Kaplan et al. (2020) and Hoffmann et al. (2022), the main objective has been guidance on how to optimally scale Large Language Models with Transformers. Follow-up work investigated the data constrained setting (Muennighoff et al., 2023), the effect of data pruning (Sorscher et al., 2022), extreme token per parameter ratios (Gadre et al., 2024). Furthermore, replication efforts regarding the scaling laws established in Kaplan et al. (2020) and Hoffmann et al. (2022) have been performed in order to reconcile their findings (Besiroglu et al., 2024; Pearce & Song, 2024; Porian et al., 2024). Critical practical considerations such as specific architectures and hyperparameters on the resulting scaling laws have been investigated (McLeish et al., 2025). The recent survey Li et al. (2025) gives a comprehensive overview and give practical guidelines in establishing scaling laws. Scaling laws have also been investigated theoretically, providing justification for the functional forms used in practice (Amari et al., 1992; Amari, 1993; Seung et al., 1992; Amari & Murata, 1993; Cortes et al., 1993; Yarotsky, 2018; Liang et al., 2020; Sharma & Kaplan, 2022; Hutter, 2021; Bahri et al., 2024).

**Incorporating inference characteristics into scaling laws.** Multiple studies seek to include inference characteristics such as the time-to-first-token (latency) and the time-per-token (throughput) into their considerations on model scaling. Sardana et al. (2024) propose to incorporate inference costs into scaling laws for an expected inference compute demand. Gadre et al. (2024) investigate scaling laws in training regimes with high token/parameter ratios, much higher than "Chinchilla-optimal", which incurs higher inference speeds due to smaller models. Bian et al. (2025) devise inference-aware scaling laws, focusing on obtaining the most inference efficient model for a certain performance. Paliotta et al. (2025) show, that under fixed time budget during inference, distilling Transformers into linear time-complexity Mamba models leads to higher performance on reasoning tasks, as their faster inference speeds allow for better scaling with inference compute.

**Other scaling behaviors.** Beyond scaling behavior with model parameters and training data, other scaling behaviors have been investigated. Hernandez et al. (2021) considers scaling laws for transfer learning. Clark et al. (2022) and Abnar et al. (2025) investigate scaling laws for routed language models, such as the widely considered Mixture-of-Experts method (Shazeer et al., 2017). Scaling inference compute is a major consideration for LLM reasoning models (OpenAI, 2024). For example Snell et al. (2025); Brown et al. (2024); Muennighoff et al. (2025) demonstrated such scaling behavior with additional inference tokens. Kumar et al. (2025) devise precision-aware scaling laws, investigating the tradeoffs between precision, parameters and data. Tao et al. (2024) suggest the vocabulary size as additional parameter when scaling language models. Busbridge et al. (2025) investigate scaling laws for distilled models based on the compute budget allocation between teacher and student. Zhao et al. (2025) reconcile the smooth improvements predicted by scaling laws with the reported sudden emergent capabilities of LLMs at scale through distributional scaling laws. Chen et al. (2025) introduce parallel scaling laws, where compute is scaled by using a single set of model parameters in parallel with different learnable input transformations and output aggregation. Related to our work, Xiong et al. (2024) and Shi et al. (2025) investigate the scaling behavior of transformer models w.r.t. their context length. Springer et al. (2025) show that overtrained models are harder to fine-tune.

Closest to our work are Shen et al. (2024) and Poli et al. (2024). Shen et al. (2024) demonstrate scaling behavior of their considered linear time-complexity architectures that is on par with Transformers. Poli et al. (2024) shows, that hybrids between linear time-complexity and transformer models can

improve upon Transformers. Contrary, our work shows that the xLSTM linear time-complexity architecture outscales Transformers for language modeling.

## 6 LIMITATIONS AND FUTURE WORK

The main focus of this work is a comparative study of the training scaling behavior of Transformer and xLSTM architectures in terms of cross-entropy loss. We do not consider the impact of different training data distributions, nor do we investigate scaling behavior on other downstream tasks; instead, we build on the findings of related work on these aspects (Sardana et al., 2024; Gadre et al., 2024; Porian et al., 2024). Similarly, our empirical inference runtime scaling is designed to capture the fundamental differences in computational complexity with respect to sequence length between Transformers and xLSTM. Therefore, we adopt a fair and controlled comparative setup, focusing on single-GPU experiments rather than exhaustive inference optimizations.

Future work could extend the scaling comparisons to Mixture-of-Expert or hybrid architectures combining attention and xLSTM, explore diverse data distributions, include additional downstream and long-context tasks, and investigate inference runtimes in production scale multi-GPU regimes to provide further insights into efficient sequence modeling.

## 7 CONCLUSION

Our study provides a systematic comparison of scaling behaviors between xLSTM and Transformer architectures. We show that xLSTMs are Pareto-dominant in training loss versus compute, maintain consistent power-law exponents in the overtraining regime, and scale more efficiently with context length due to their linear complexity. While our results suggest a universal relationship between performance and model size that applies to both compute-optimal Transformers and xLSTM models, we find that compute-optimal xLSTM models are larger than their Transformer counterparts and that the compute-optimal model size of xLSTMs is robust to variations in context length. During inference, xLSTM models achieve lower time to first tokens and generation step times than Transformer models of the same size. These results are well explained by our runtime model, which is grounded in theoretical FLOP and memory operation calculations and shows close agreement with the empirical data. Throughout all experiments, we find that the advantages of xLSTM grow with context length, both for training and inference characteristics, positioning xLSTM as a promising and scalable architecture for future language models.

## REPRODUCIBILITY STATEMENT

We release the code to reproduce our experiments, the datasets of training runs as well as results for inference publicly upon acceptance to facilitate future research in this direction. The datasets of training runs have been obtained using the publicly available xLSTM 7B training repository (`https://github.com/NX-AI/xlstm-jax`) using the model configurations stated in Appendix D. Inference results have been obtained using the publicly available benchmarking pipeline for efficient xLSTM kernels (`https://github.com/NX-AI/mlstm_kernels`), more specifically, the model benchmarks, not those for individual kernels.

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

APPENDIX

CONTENTS

# A    EXTENDED TRAINING SCALING BEHAVIOR

## A.1    DETAILS ON THE EXPERIMENTAL SETUP

We provide additional details on our experiments, that we conducted on a cluster of NVIDIA H100 GPUs.

**Model Configurations.** In Appendix D we provide a list of model architecture configurations for all Transformer and xLSTM models used in our scaling law study in Token/Param (App. D.1) and IsoFLOP (App. D.2) training setups.

**General Hyperparameters.** We use the AdamW optimizer with $\beta_1 = 0.99$, $\beta_2 = 0.95$, $\epsilon = 10^{-8}$, weight decay $0.1$ and gradient clipping norm $0.5$. Our learning rate schedule comprises three stages: A linear warm-up of $750$ training steps, a cosine decay to $10\%$ of the peak learning rate and a final linear cool-down of $1000$ training steps. While we keep the steps for warm-up and cool-down constant, we match length of our learning rate decay to the token budget, which is either determined by a specific token-to-parameter ratio or a compute budget for a given model size (see Sec. 3.1). Unless specified otherwise, we use a context length of $8192$ for our scaling law study.

**Hyperparameters for Token/Param setup.** We specify our batch sizes and learning rates for our experiments in the overtraining regime with large token-to-parameter ratios for xLSTM and Transformer models in Tab. 19 and 20, respectively. For larger models we decrease the learning rate and use larger batch sizes. We find that for very large token-to-parameter ratios the performance in terms of validation loss becomes less sensitive to the choice of learning rate.

**Hyperparameters for IsoFLOP setup.** For our IsoFLOP experiments we use a batch size of 1M tokens for all but the largest compute budget of 6e+20 FLOPs, where we double the batch size to 2M tokens, as the training runs would become prohibitively long (see Tab. 1). In contrast to the Token/Param experiments, we do not increase the batch size with model size, since we found that this leads to loss offsets in the isoflop profiles (see Fig. 7, left). Instead, we keep the batch size constant for each compute budget, regardless of the model size. We validate this choice by repeating the experiments for the isoflop profile with compute budget 1e+20 with a batch size of 1M and 2M tokens. We find that the larger batch size yields a higher validation loss due to fewer training steps, but does not have a major impact on the optimal number of parameters $N^*$ for this compute budget (see Fig. 7, right). Starting from the Token/Param learning rates, we tune the learning rates for selected model sizes, and use the best learning rates for models of similar size.

Table 1: Batch sizes used for the IsoFLOP training setup at context length $T = 8192$. For the other context lengths $T$ we adjust $B$ such that batch size in number of tokens $B \times T$ remains constant.

| IsoFLOP | $B$ (seqs) | $B \times T$ (tokens) |
|---|---|---|
| 6e+18 | 128 | 1,048,576 |
| 1e+19 | 128 | 1,048,576 |
| 3e+19 | 128 | 1,048,576 |
| 1e+20 | 128 | 1,048,576 |
| 6e+20 | 256 | 2,097,152 |

## A.2    DETAILS ON THE PARAMETRIC LOSS SURFACE FIT

For the parametric loss surface fit $\hat{L}(N, D)$ in Figure 1 we follow the procedure outlined in Busbridge et al. (2025, App. F.1). We fit the coefficients $\{E, A, B, \alpha, \beta, \gamma\}$ for the parametric function of the loss surface $\hat{L}(N, D)$ in (2) with different values for the Huber $\delta$. Similar to Busbridge et al. (2025), we observe that including $\gamma$, significantly improves the quality of our fits (see Fig. 8. We use the the Token/Param training configurations for Transformer (31 samples) and xLSTM (35 samples) from our dataset of training runs and fit over a grid of L-BFGS-B initializations given by: $\log A \in \{0.0, 5.0, 10.0, 15.0, 20.0\}$, $\log B \in \{0.0, 5.0, 10.0, 15.0, 20.0\}$, $\log E \in \{-1.0, -0.5, 0.0, 0.5, 1.0\}$, $\alpha \in \{0.0, 0.2, 0.5, 1.0\}$, $\beta \in \{0.0, 0.2, 0.5, 1.0\}$ and $\gamma \in \{0.0, 0.5, 1.0, 1.5\}$.

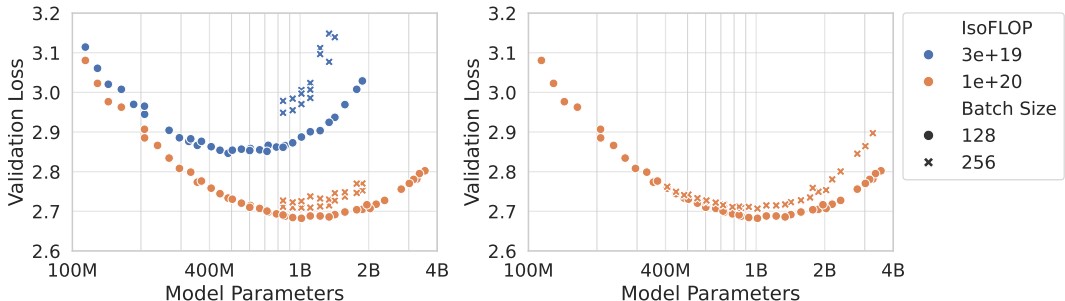

Figure 7: Impact of the batch size on IsoFLOP profiles. **Left:** IsoFLOP curves with large batch size and different learning rates for large models. Varying the batch size for different model sizes, leads to offsets in the IsoFLOP profile, which are more pronounced for smaller compute budgets. **Right:** IsoFLOP profile for compute budget 1e+20 with different batch sizes. The larger batch size leads to larger loss, but similar optimal model size.

In Tab. 2, we report the coefficients that achieve the lowest MSE on the fit data out of all initializations for different Huber $\delta$. We find that the optimal fit parameters are sensitive to the choice of $\delta$. For $\delta \geqslant 0.1$ the optimal values for the fit parameters did not change in the digits shown in Tab. 2.

Table 2: Optimal fit parameters for the loss surface $\hat{L}(N, D)$ model from equation (2) for Transformer and xLSTM models for different Huber $\delta$. In Figure 1 we plot the fit for $\delta = 10^{-3}$.

|  | Huber $\delta$ | $\log A$ | $\log B$ | $\log E$ | $\alpha$ | $\beta$ | $\gamma$ |
|---|---|---|---|---|---|---|---|
| Transformer | $10^{-5}$ | 12.96 | 14.35 | 0.05 | 0.58 | 0.55 | 0.28 |
|  | $10^{-3}$ | 11.99 | 13.35 | 0.01 | 0.53 | 0.51 | 0.29 |
|  | $\geqslant 10^{-1}$ | 14.45 | 16.33 | 0.09 | 0.64 | 0.63 | 0.25 |
| xLSTM | $10^{-5}$ | 16.13 | 17.10 | 0.07 | 0.71 | 0.66 | 0.24 |
|  | $10^{-3}$ | 16.22 | 17.31 | 0.11 | 0.73 | 0.67 | 0.24 |
|  | $\geqslant 10^{-1}$ | 15.46 | 16.53 | 0.18 | 0.71 | 0.65 | 0.26 |

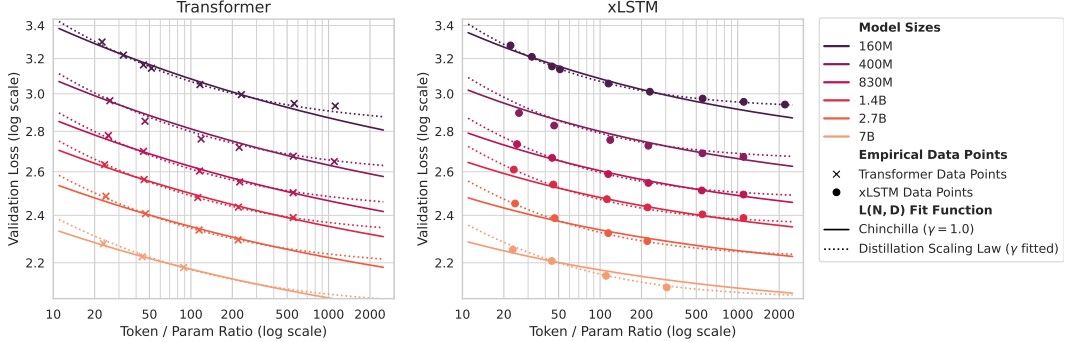

Figure 8: Comparison between the parametric fit with $\gamma = 1$ (Hoffmann et al., 2022) and $\gamma$ as free parameter (Busbridge et al., 2025). Including $\gamma$ as fit parameter improves the fit quality.

### A.3 POWER-LAW EXPONENTS IN OVER-TRAINING

In Tab. 3 we report the power-law exponents for different token-to-parameter ratios.

Table 3: Power-law exponents $\eta$ for increasing token-to-parameter ratios $M$.

| $M$ | Transformer | xLSTM |
|---|---|---|
| 22 | 0.050 | 0.047 |
| 44 | 0.048 | 0.046 |
| 110 | 0.047 | 0.046 |
| 220 | 0.048 | 0.047 |
| 550 | 0.049 | 0.047 |
| 1100 | - | 0.047 |

### A.4 ADDITIONAL RESULTS: ISOFLOP APPROACH

**Comparison of our scaling law to Porian et al. (2024).** In order to validate our scaling law framework, we compare our power-law fits for the optimal model size from Fig. 4 with the results from Porian et al. (2024). Porian et al. (2024) investigate and resolve the discrepancies in scaling laws between the influential works by Kaplan et al. (2020) and Hoffmann et al. (2022). We find that our power-law coefficient $a_{\mathrm{ours}} = 0.575$ is very close to the coefficient reported in Figure 1d) from Porian et al. (2024) with $a_{\mathrm{Porian,d}} = 0.571$ and even falls well into their confidence interval of $(0.56, 0.59)$, despite the well-documented reproducibility challenges in scaling laws (Porian et al., 2024; Li et al., 2025; McLeish et al., 2025). Porian et al. (2024) report that for their $a_{\mathrm{Porian,d}}$ they match their learning rate cosine decay schedule to each token budget – a practice that we follow in our experimental setup (see App. A.1. This agreement validates our framework and affirms its credibility. As the final step, to fully match the coefficients reported by Hoffmann et al. (2022), Porian et al. (2024) report that it is necessary to tune learning rate, batch size and AdamW $\beta_2$ parameter individually for each model size. However, in our case this would require considerably more compute resources due to our much larger compute budgets (6e+18 - 6e+20), and hence larger model sizes used for our scaling law study.

**Compute-optimal dataset size.** In the main paper (Sec. 3.4, Fig. 4), we presented results for the compute-optimal model size. In Fig. 9 we present results w.r.t. the number of training tokens. We observe that compute-optimal xLSTMs and Transformers are trained on a similar number of tokens.

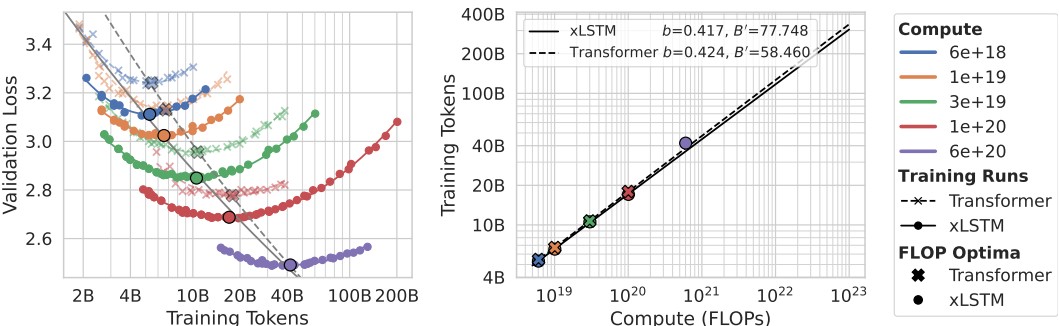

Figure 9: Left: IsoFLOP curves for varying number of training tokens with a marker at the minimum of the fit. Right: Plot of the power-law fit for the compute optimal number of training tokens $D^*(C)$. Colors indicate compute budget and marker types indicate the model types.

## A.5 ADDITIONAL RESULTS: ISOFLOP APPROACH FOR DIFFERENT CONTEXT LENGTHS

Complementary to the IsoFLOP results in Sec 3.5, where we showed scaling behavior w.r.t. the model parameters, we also show the scaling behavior w.r.t. the dataset size. The results are provided in Figure 10, showing that for xLSTM it slightly increases with context length, whereas for Transformer it substantially decreases. This is caused by the quadratic cost of the attention mechanism that becomes dominant at larger context lenghts, causing substantial compute that shifts compute-optimal models towards smaller models that are trained on less tokens. For all considered context lenghts, it is favorable to train an xLSTM model compared to a Transformer model under the same compute budget. The longer the training context length, the more favorable it is to train an xLSTM compared to a Transformer.

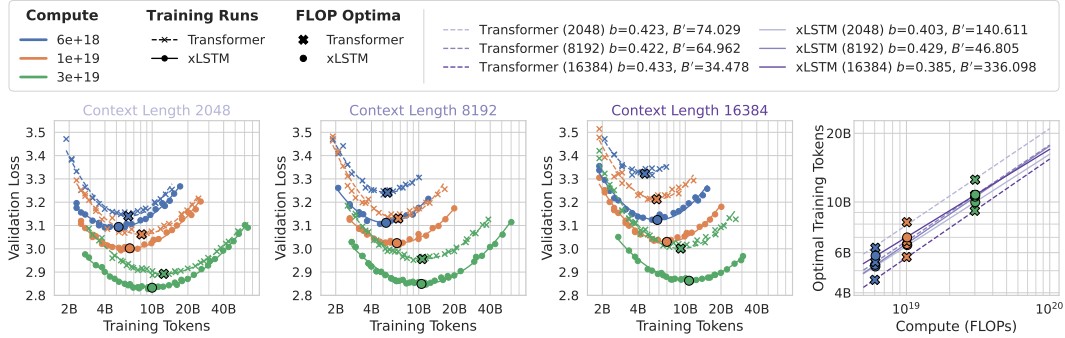

Figure 10: IsoFLOP curves for xLSTM and Transformer for different context lengts and varying number of training tokens.

By rearranging the data obtained from the IsoFLOP approach under different context lengths, one can also fit scaling laws for the context length. This is done equivalently to scaling laws for the model parameters and number of training tokens (Eq. (3)). Figure 11 shows the results w.r.t. the number of model parameters and Figure 12 shows the results w.r.t. the number of training tokens. The obtained scaling laws mirror the findings from before. Compute-optimal xLSTM models have more or less constant model size and use slightly more tokens w.r.t. the context length. Compute-optimal Transformer models are becoming smaller and use less training tokens w.r.t. the context length.

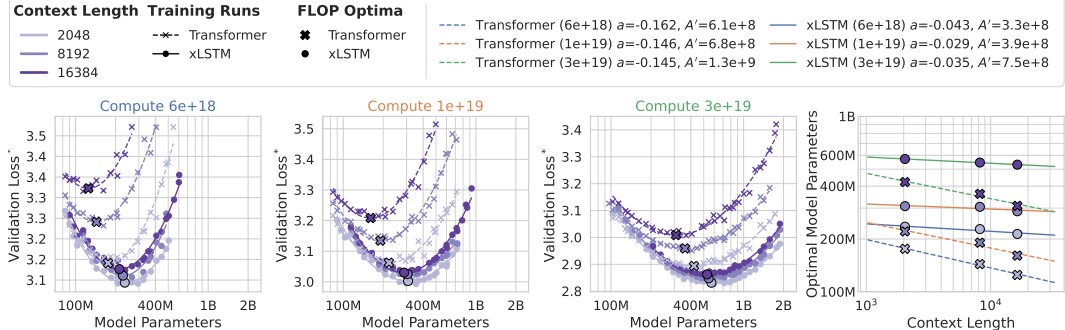

Figure 11: Left: IsoFLOP curves for xLSTM and Llama as a function of model parameters at 3 different compute budgets. Right: Plot of the power-law fits for the compute optimal number of parameters dependent on the context length $N^*(T)$. Colors indicate context length and marker types indicate the model types.

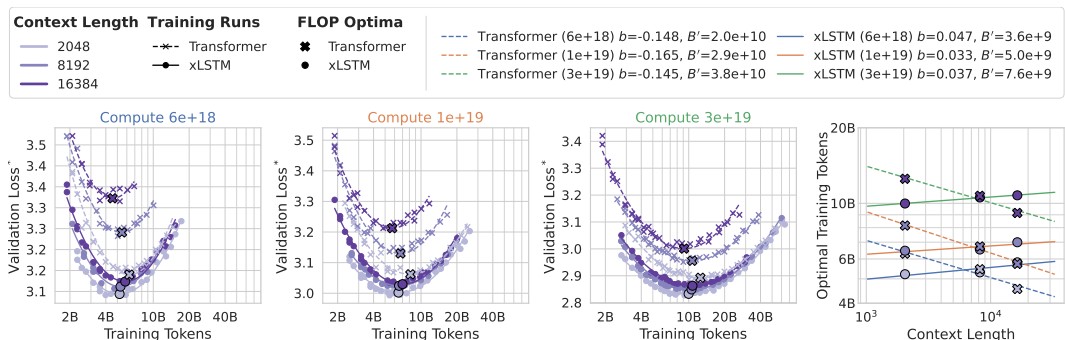

Figure 12: Left: IsoFLOP curves for xLSTM and Llama as a function of training token at 3 different compute budgets. Right: Plot of the power-law fits for the compute optimal number of parameters dependent on the context length $N^*(T)$. Colors indicate context length and marker types indicate the model types.

# B ACCOUNTING: PARAMETERS, CACHE SIZES, FLOPS, MEMORY OPERATIONS

In this section, we count number of parameters (App. B.1), memory state or KV cache size B.2, FLOPs (App. B.3), and memory operations (App. B.4) for mLSTM models based on the architecture of xLSTM 7B (Beck et al., 2025b) and Transformer models with Self-Attention based on the Llama 3 architecture (Grattafiori et al., 2024).

We use the notation defined in Tab. 4.

We start with counting the number of memory operations and FLOPs for matrix multiplication, which is a very common operation in neural networks. A linear layer with input $X$ and output $Y$ and weight matrix $W$ can be written as

$$\underset{(B \times d_{\text{out}})}{Y} = \underset{(B \times d_{\text{in}})}{X} \underset{(d_{\text{in}} \times d_{\text{out}})}{W^{\top}}. \tag{5}$$

This linear layer has $2Bd_{\text{in}}d_{\text{out}}$ FLOPs:

$$\text{FLOPs}_{\text{linear}} = 2Bd_{\text{in}}d_{\text{out}} \tag{6}$$

In order to compute the output $Y$, we need to read the input $X$ and the weights $W$ and write the output $Y$. This yields

$$\text{Bytes}_{\text{linear}} = B(d_{\text{in}} + d_{\text{out}}) \times \text{bytes}_{\text{XY}} + d_{\text{in}}d_{\text{out}} \times \text{bytes}_{\text{W}} \tag{7}$$

memory operations in loaded and stored bytes. We will use these counts throughout the remainder of this section.

Table 4: **Notation** for FLOP and Memory Operation Counts.

| Symbol | Description |
|---|---|
| $B$ | Batch size |
| $T, (T_{\text{p}}, T_{\text{g}})$ | Sequence length, (prefill length, generation length) |
| $S$ | Query sequence length (only for Self-Attention) |
| $L$ | Chunk size |
| $d_{\text{hv}}$ | Head dimension for values and hidden states |
| $d_{\text{qk}}$ | Head dimension for queries and keys |
| $d_{\text{model}}$ | Model / Embedding dimension |
| $d_{\text{ff}}$ | Feedforward dimension |
| $p_{\text{ff}}$ | Feedforward projection factor |
| $p_{qk}$ | Query key projection factor |
| $n_{\text{head}(,q)}$ | Number of (query) heads |
| $n_{\text{head},kv}$ | Number of key and value heads |
| $n_{\text{chunk}}$ | Number of chunks |
| $n_{\text{vocab}}$ | Vocabulary size |
| $n_{\text{layer}}$ | Number of layers |
| $F_{\text{OP}}$ | FLOPs for the operation OP (e.g. $\exp$) |
| $F_{\text{causal}}$ | Factor that accounts for causality, typically 0.5 |
| $\text{bytes}_{\text{X}}$ | Number of bytes used for each element in tensor X |

### B.1 PARAMETER COUNTS

We count the number of parameters of mLSTM models (B.1.1) and Transformer models (B.1.2). We include embedding and normalization layer parameters in our parameter counts.

#### B.1.1 MLSTM PARAMS

For the mLSTM models, we use the optmized xLSTM architecture from Beck et al. (2025b) and count the parameters in Tab. 5.

Table 5: **Parameter counts** for the **mLSTM Model**.

| Parameters | |
|---|---|
| **Embeddings:** | $n_{\text{vocab}} d_{\text{model}}$ |
| *mLSTM (single layer)* | |
| **PreNorm:** | $d_{\text{model}}$ |
| **QKV:** | $d_{\text{model}} n_{\text{head}} (2 d_{\text{qk}} + d_{\text{hv}})$ |
| **Inpute & Forget Gates:** | $2 d_{\text{model}} n_{\text{head}} + 2 n_{\text{head}}$ |
| **Output Gate:** | $d_{\text{model}} n_{\text{head}} d_{\text{hv}}$ |
| **Output Norm:** | $n_{\text{head}} d_{\text{hv}}$ |
| **Output Projection:** | $d_{\text{model}} n_{\text{head}} d_{\text{hv}}$ |
| *Total mLSTM layer $N_{\text{mLSTM,layer}}$:* | $d_{\text{model}} n_{\text{head}} (2 d_{\text{qk}} + d_{\text{hv}} + 2) + 2 d_{\text{model}}^2 + 2 n_{\text{head}} + 2 d_{\text{model}}$ |
| *Feedforward (single layer)* | |
| **PreNorm:** | $d_{\text{model}}$ |
| **MLPs:** | $3 d_{\text{model}} d_{\text{ff}}$ |
| *Total Feedforward $N_{\text{ff,layer}}$:* | $3 d_{\text{model}} d_{\text{ff}} + d_{\text{model}}$ |
| **Output Norm:** | $d_{\text{model}}$ |
| **Unembedding:** | $d_{\text{model}} n_{\text{vocab}}$ |
| **Total mLSTM model $N_{\text{mLSTM}}$:** | $n_{\text{layer}} (N_{\text{mLSTM,layer}} + N_{\text{ff,layer}}) + 2 d_{\text{model}} n_{\text{vocab}} + d_{\text{model}}$ |

#### B.1.2 TRANSFORMER PARAMS

For the Transformer models, we assume the Llama architecture with Grouped-Query Attention from Grattafiori et al. (2024) and count the parameters in Tab. 6.

### B.2 MEMORY STATE AND KV-CACHE SIZE

In Tab. 7 we list the memory state and KV cache sizes for the mLSTM and Transformer model architectures. We compare the mLSTM with standard Multi-Head Attention (MHA) (Vaswani et al., 2017), Grouped-Query Attention (GQA) (Ainslie et al., 2023) and Multi-Head Latent Attention (DeepSeek-AI, 2024a).

In contrast to the KV caches of the attention variants, the mLSTM has a fixed size memory state that does not depend on the sequence length $T$.

We compare the size of the memory state and KV cache sizes in number of elements. To obtain the number of bytes, we multiply by number of bytes per element $\text{bytes}_{\text{X}}$.

Table 6: **Parameter counts** for the **Transformer Self-Attention Model**.

| Parameters | |
|---|:---:|
| **Embeddings:** | $n_{\text{vocab}}d_{\text{model}}$ |
| *Self-Attention (single layer)* | |
| **PreNorm:** | $d_{\text{model}}$ |
| **QKV:** | $d_{\text{model}}\big(d_{\text{qk}}n_{\text{head,q}} + (d_{\text{qk}} + d_{\text{hv}})n_{\text{head,kv}}\big)$ |
| **Output Projection:** | $d_{\text{model}}n_{\text{head,q}}d_{\text{hv}}$ |
| *Total Attention layer $N_{\text{Att,layer}}$:* | $d_{\text{model}}(d_{\text{qk}}n_{\text{head,q}} + d_{\text{qk}}n_{\text{head,kv}} + d_{\text{hv}}n_{\text{head,kv}}) + d_{\text{model}}^2 + d_{\text{model}}$ |
| *Feedforward (single layer)* | |
| **PreNorm:** | $d_{\text{model}}$ |
| **MLPs:** | $3d_{\text{model}}d_{\text{ff}}$ |
| *Total Feedforward $N_{\text{ff,layer}}$:* | $3d_{\text{model}}d_{\text{ff}} + d_{\text{model}}$ |
| **Output Norm:** | $d_{\text{model}}$ |
| **Unembedding:** | $d_{\text{model}}n_{\text{vocab}}$ |
| **Total Transformer model $N_{\text{Att}}$:** | $n_{\text{layer}}(N_{\text{Att,layer}} + N_{\text{ff,layer}}) + 2d_{\text{model}}n_{\text{vocab}} + d_{\text{model}}$ |

Table 7: **Memory State and KV-Cache Sizes** for different **Sequence-Mix operations**. All terms denote the number of elements.

| Sequence Mix Operation | Memory Size in #Elements |
|---|:---:|
| Multi-Head Attention (MHA): | $2n_{\text{head,q}}d_{\text{hv}}T$ |
| Grouped-Query Attention (GQA): | $2n_{\text{head,kv}}d_{\text{hv}}T$ |
| Multi-Head Latend Attention (MLA): | $\frac{9}{2}d_{\text{hv}}T$ |
| mLSTM: | $n_{\text{head,q}}(d_{\text{hv}}d_{\text{qk}} + d_{\text{qk}} + 1)$ |

## B.3 FLOP COUNTS

In this section, we count the FLOPs for the mLSTM and the Transformer model architecture. For each model architecture we count the sequence length dependent FLOPs for the sequence mix layer first, i.e. the mLSTM cell (B.3.1) and the Self-Attention layer (B.3.3), and then combine them with the FLOPs of the other layers in the model architecture to obtain the total FLOPs for the mLSTM (B.3.2) and the Transformer model (B.3.4).

We do not drop subleading terms and set also count FLOPs for all operations equally, i.e. $F_{\text{OP}} = 1$. We also count the FLOPs for the normalization layers with $F_{\text{norm}} = 3$ (we assume the factor of 3 because we have mean, variance and division operations). The skip connection FLOPs are counted with $F_{\text{skip}} = 1$, or if neglected with $F_{\text{skip}} = 0$. Following our training configuration, we use the chunkwise-parallel formulation with chunk size $L = 64$ and $F_{\text{causal}} = 0.5$ for the FLOP counts and scaling laws in the main text.

### B.3.1 MLSTM CELL FLOPS

The mLSTM is a linear RNN with gating and can be computed either with a recurrent, a fully parallel or a chunkwise-parallel formulation (Beck et al., 2025a). Each of these formulations has a different FLOP and memory operation count. For training and for prefill in inference the mLSTM relies on the chunkwise-parallel formulation, which parallelizes the computation over the input sequence and can therefore fully utilize modern hardware. For generation, the mLSTM uses the recurrent formulation, which uses constant compute and memory per generation step (i.e. compute and memory requirements are independent of the sequence length).

In this section, we count the number of FLOPs for both the chunkwise-parallel and the recurrent formulation of the mLSTM cell.

**Chunkwise-Parallel Formulation (Tab. 8, Eq. 8).** We list the FLOP counts for the individual terms of the chunkwise-parallel mLSTM formulation for a single head and a single chunk in Tab. 8.

To obtain the total FLOPs for a full sequence of length $T$, we multiply these counts by the number of (query) heads $n_{\text{head}}$ and chunks $n_{\text{chunk}} = T/L$. This yields

$$
\begin{aligned}
\text{FLOPs}_{\text{mLSTM,cwp}} = n_{\text{head}} \times \Bigg( & TLF_{\text{causal}}\left(2(d_{\text{qk}} + d_{\text{hv}}) + 8\right) + TL \\
& + 2TF_{\text{causal}} + T\left(4d_{\text{qk}}d_{\text{hv}} + 6d_{\text{qk}} + 4d_{\text{hv}} + 13\right) \\
& + \frac{T}{L}\left(2d_{\text{qk}}d_{\text{hv}} + 2d_{\text{qk}} + 5\right) \Bigg).
\end{aligned}
\tag{8}
$$

Table 8: **FLOP counts** for the **chunkwise-parallel mLSTM formulation** for mLSTM. All terms denote the FLOP count per head and chunk.

| FLOPs | Exact | Simplified ($F_{\text{OP}} = 1$) |
|---|:---:|:---:|
| *Recurrent computation of the inter chunk states* | | |
| **Gates:** | $2L + \frac{1}{2}L(L+1)$ $+L(1 + F_{\text{exp}} + F_{\text{log}} + F_{\text{sig}}) + 3 + F_{\text{max}} + F_{\text{exp}}$ | $0.5L^2 + 6.5L + 5$ |
| **Numerator:** | $2d_{\text{qk}}d_{\text{hv}} + 2Ld_{\text{qk}}d_{\text{hv}} + Ld_{\text{qk}}$ | $2d_{\text{qk}}d_{\text{hv}} + 2Ld_{\text{qk}}d_{\text{hv}} + Ld_{\text{qk}}$ |
| **Denominator:** | $2d_{\text{qk}} + 2Ld_{\text{qk}}$ | $2d_{\text{qk}} + 2Ld_{\text{qk}}$ |
| *Parallel computation of the intra chunk outputs* | | |
| **Cumulative Forget Gates:** | $\frac{1}{2}L(L+1) + L(F_{\text{log}} + F_{\text{sig}})$ | $0.5L^2 + 2.5L$ |
| **Gate Matrix:** | $F_{\text{causal}} \times \left(L^2(3 + F_{\text{exp}} + F_{\text{max}}) + L(1 + F_{\text{max}})\right)$ | $F_{\text{causal}} \times \left(5L^2 + 2L\right)$ |
| **Intra Outputs:** | $F_{\text{causal}} \times \left(2L^2(d_{\text{qk}} + d_{\text{hv}}) + 3L^2\right)$ | $F_{\text{causal}} \times \left(2L^2(d_{\text{qk}} + d_{\text{hv}}) + 3L^2\right)$ |
| *Parallel computation of the inter chunk outputs* | | |
| **Inter Outputs:** | $2Ld_{\text{qk}}d_{\text{hv}} + 3Ld_{\text{qk}}$ | $2Ld_{\text{qk}}d_{\text{hv}} + 3Ld_{\text{qk}}$ |
| *Combination of inter and intra chunk outputs* | | |
| **Output Combination:** | $2Ld_{\text{hv}} + L(1 + F_{\text{max}} + F_{\text{abs}} + F_{\text{exp}})$ | $2Ld_{\text{hv}} + 4L$ |
| **Total:** | — | $L^2 F_{\text{causal}}\left(2(d_{\text{qk}} + d_{\text{hv}}) + 8\right) + L^2 + 2LF_{\text{causal}}$ $+L\left(4d_{\text{qk}}d_{\text{hv}} + 6d_{\text{qk}} + 4d_{\text{hv}} + 13\right)$ $+ \left(2d_{\text{qk}}d_{\text{hv}} + 2d_{\text{qk}} + 5\right)$ |

**Recurrent Formulation (Tab. 9, Eq. 9).** We list the FLOP counts for the individual terms of the recurrent mLSTM formulation for a single head and a single time step in Tab. 9.

To obtain the total counts for one generation step, we multiply by the number of heads $n_{\text{head}}$. This yields

$$
\text{FLOPs}_{\text{mLSTM,rec}} = n_{\text{head}} \times \left(6d_{\text{qk}}d_{\text{hv}} + 7d_{\text{qk}} + d_{\text{hv}} + 12\right).
\tag{9}
$$

Table 9: **FLOP counts** for the **recurrent mLSTM formulation** for mLSTM. All terms denote the FLOP count for a single timestep per head.

| FLOPs | Exact | Simplified ($F_{\text{OP}} = 1$) |
|---|:---:|:---:|
| **Gates:** | $4 + 2F_{\text{exp}} + F_{\text{log}} + F_{\text{sig}} + F_{\text{max}}$ | $9$ |
| **Memory Cell Update:** | $4d_{\text{qk}}d_{\text{hv}}$ | $4d_{\text{qk}}d_{\text{hv}}$ |
| **Denominator & Scale:** | $6d_{\text{qk}} + d_{\text{hv}} + 1 + F_{\text{abs}} + F_{\text{max}}$ | $6d_{\text{qk}} + d_{\text{hv}} + 3$ |
| **Output:** | $2d_{\text{hv}}d_{\text{qk}} + d_{\text{qk}}$ | $2d_{\text{hv}}d_{\text{qk}} + d_{\text{qk}}$ |
| **Total:** | — | $6d_{\text{qk}}d_{\text{hv}} + 7d_{\text{qk}} + d_{\text{hv}} + 12$ |

### B.3.2 mLSTM Model FLOPs

The number of FLOPs for the backbone is identical for training, prefill and generation as the operations (embeddings, linear layers and layernorms) do not depend on the sequence length. Therefore, we count the FLOPs per token for the mLSTM backbone. To obtain the total FLOPs for the specific setting we have to use the respective expression for the mLSTM cell FLOPs from Appendix B.3.1.

**mLSTM Backbone (Tab. 10).** We count the FLOPs for the mLSTM backbone for a single token in Tab. 10 and leave the mLSTM cell FLOPs unspecified. The number of tokens for one batch of sequences is $BT$.

Table 10: **FLOP counts** for the **mLSTM backbone**. All terms denote the FLOP count per token, i.e. to obtain the FLOPs for one batch we multiply by $BT$ tokens.

| FLOPs | |
|---|---|
| **Embeddings:** | — |
| *mLSTM (single layer)* | |
| **PreNorm & Skip:** | $d_{\text{model}}(F_{\text{skip}} + F_{\text{norm}})$ |
| **QKV:** | $2d_{\text{model}}n_{\text{head}}(2d_{\text{qk}} + d_{\text{hv}})$ |
| **Inpute & Forget Gates:** | $2d_{\text{model}}n_{\text{head}} + 2n_{\text{head}}$ |
| **mLSTM Cell:** | $\text{FLOPs}_{\text{mLSTM}}$ |
| **Output Gate:** | $2d_{\text{model}}n_{\text{head}}d_{\text{hv}} + n_{\text{head}}d_{\text{hv}}F_{\text{sig}}$ |
| **Output Norm:** | $n_{\text{head}}d_{\text{hv}}F_{\text{norm}}$ |
| **Output Projection:** | $2d_{\text{model}}n_{\text{head}}d_{\text{hv}}$ |
| *Total mLSTM layer* $\text{FLOPs}_{\text{mLSTM,layer}}$: | — |
| *Feedforward (single layer)* | |
| **PreNorm & Skip:** | $d_{\text{model}}(F_{\text{skip}} + F_{\text{norm}})$ |
| **MLPs:** | $6d_{\text{model}}d_{\text{ff}}$ |
| **Activations:** | $d_{\text{ff}}(1 + F_{\text{swish}})$ |
| *Total Feedforward* $\text{FLOPs}_{\text{ff,layer}}$: | — |
| **Output Norm:** | $d_{\text{model}}F_{\text{norm}}$ |
| **Unembedding:** | $2d_{\text{model}}n_{\text{vocab}}$ |
| **Total mLSTM model** $\text{FLOPs}_{\text{mLSTM,model}}$: | — |

### B.3.3 SELF-ATTENTION FLOPS

We count the FLOPs for a single Self-Attention head during training or prefill and generation in Tab. 11. We denote the number of keys and values in the sequence as $T$, and the number of queries as $S$. During prefill we have $S = T$, since the input sequence is processed in parallel and during autoregressive generation we have $S = 1$, since we generate one token at a time. We typically use $F_{\text{softmax}} = 5$ and $F_{\text{causal}} = 0.5$ following Busbridge et al. (2025) as FLOP factor for softmax (sm).

Table 11: **FLOP counts** for **Self-Attention**. All terms denote the FLOP count per (query) head.

| FLOPs | Generic | Prefill ($S = T$) | Generate ($S = 1$) |
|---|---|---|---|
| *Attention computation* | | | |
| **Logits:** | $2STd_{\text{qk}} \times F_{\text{causal}}$ | $2T^2d_{\text{qk}} \times F_{\text{causal}}$ | $2Td_{\text{qk}} \times F_{\text{causal}}$ |
| **Attention:** | $STF_{\text{softmax}} \times F_{\text{causal}}$ | $T^2F_{\text{softmax}} \times F_{\text{causal}}$ | $TF_{\text{softmax}} \times F_{\text{causal}}$ |
| **Hiddens/Outputs:** | $2STd_{\text{hv}} \times F_{\text{causal}}$ | $2T^2d_{\text{hv}} \times F_{\text{causal}}$ | $2Td_{\text{hv}} \times F_{\text{causal}}$ |
| **Total:** | $2STF_{\text{causal}}(d_{\text{qk}} + d_{\text{hv}} + 0.5F_{\text{sm}})$ | $2T^2F_{\text{causal}}(d_{\text{qk}} + d_{\text{hv}} + 0.5F_{\text{sm}})$ | $2TF_{\text{causal}}(d_{\text{qk}} + d_{\text{hv}} + 0.5F_{\text{sm}})$ |

**Self-Attention in Training (forward only) and Prefill (Eq. 10).** To obtain the FLOPs for all Self-Attention heads for a full sequence $T$ or $T_{\text{p}}$, we multiply by the number of (query) heads $n_{\text{head,q}}$ and the number of tokens $T$. This yields

$$\text{FLOPs}_{\text{Att,train-pref}} = 2F_{\text{causal}}T^2n_{\text{head,q}}(d_{\text{qk}} + d_{\text{hv}} + 0.5F_{\text{sm}}). \tag{10}$$

**Self-Attention FLOPs in Generation (Eq. 16).** During generation the number of FLOPs per token is dependent on the number of previous tokens $T = T_{\text{p}} + t_{\text{g}}$, where $T_{\text{p}}$ is the number of prefill tokens and $t_{\text{g}}$ is the number of generated tokens so far. We denote the number of total tokens to be generated as $T_{\text{g}}$. To obtain the FLOP counts for the $t_{\text{g}}$-th generated token, we need to multiply the FLOPs for the Self-Attention layer by the number of (query) heads $n_{\text{head,q}}$. We obtain the FLOPs for the $t_{\text{g}}$-th generated token as

$$\text{FLOPs}_{\text{Att,gen-step}}(t_{\text{g}}) = 2F_{\text{causal}}n_{\text{head,q}}(d_{\text{qk}} + d_{\text{hv}} + 0.5F_{\text{sm}})(T_{\text{p}} + t_{\text{g}}). \tag{11}$$

With $a = 2F_{\text{causal}}n_{\text{head,q}}\left(d_{\text{qk}} + d_{\text{hv}} + 0.5F_{\text{sm}}\right)$ we can compute the total FLOPs for $T_{\text{g}}$ generated tokens as the sum of FLOPs for each generated token as

$$\text{FLOPS}_{\text{Att,gen-seq}} = \sum_{t_{\text{g}}=1}^{T_{\text{g}}} \text{FLOPS}_{\text{Att,gen-step}}(t_{\text{g}}) \tag{12}$$

$$= \sum_{t_{\text{g}}=1}^{T_{\text{g}}} (aT_{\text{p}} + at_{\text{g}}) \tag{13}$$

$$= aT_{\text{p}}T_{\text{g}} + a\sum_{t_{\text{g}}=1}^{T_{\text{g}}} t_{\text{g}} \tag{14}$$

$$= aT_{\text{p}}T_{\text{g}} + \frac{1}{2}aT_{\text{g}}(T_{\text{g}} + 1). \tag{15}$$

As a result we obtain the total FLOPs with a prefill or prompt length $T_{\text{p}}$ and a total number of generated tokens $T_{\text{g}}$ as

$$\text{FLOPS}_{\text{Att,gen-seq}} = 2F_{\text{causal}}n_{\text{head,q}}\left(d_{\text{qk}} + d_{\text{hv}} + 0.5F_{\text{sm}}\right)\left(T_{\text{p}}T_{\text{g}} + \frac{1}{2}T_{\text{g}}(T_{\text{g}} + 1)\right). \tag{16}$$

### B.3.4 TRANSFORMER MODEL FLOPS

Similar to the mLSTM backbone in Appendix B.3.2, the number of FLOPs for the Transformer backbone is identical for training, prefill and generation as the operations (embeddings, linear layers and layernorms) do not depend on the sequence length. Therefore, we count the FLOPs per token for the Transformer backbone. To obtain the total FLOPs for the specific setting we have to use the respective expression for the Self-Attention layer FLOPs from Appendix B.3.3.

**Transformer Backbone (Tab. 12).** We count the FLOPs for the Transformer backbone for a single token in Tab. 12 and leave the Self-Attention FLOPs unspecified. The number of tokens for one batch of sequences is $BT$.

Table 12: **FLOP counts** for the **Transformer backbone**. All terms denote the FLOP count per token, i.e. to obtain the FLOPs for one batch we multiply by $BT$ tokens.

| FLOPs | |
|---|---|
| **Embeddings:** | — |
| *Attention (single layer)* | |
| **PreNorm & Skip:** | $d_{\text{model}}(F_{\text{skip}} + F_{\text{norm}})$ |
| **QKV:** | $2d_{\text{model}}(d_{\text{qk}}n_{\text{head,q}} + d_{\text{qk}}n_{\text{head,kv}} + d_{\text{hv}}n_{\text{head,kv}})$ |
| **Attention:** | $\text{FLOPS}_{\text{Att}}$ |
| **Output Projection:** | $2d_{\text{model}}n_{\text{head,q}}d_{\text{hv}}$ |
| *Total Attention layer* $\text{FLOPS}_{\text{Att,layer}}$: | — |
| *Feedforward (single layer)* | |
| **PreNorm & Skip:** | $d_{\text{model}}(F_{\text{skip}} + F_{\text{norm}})$ |
| **MLPs:** | $6d_{\text{model}}d_{\text{ff}}$ |
| **Activations:** | $d_{\text{ff}}(1 + F_{\text{swish}})$ |
| *Total Feedforward* $\text{FLOPS}_{\text{ff,layer}}$: | — |
| **Output Norm:** | $d_{\text{model}}F_{\text{norm}}$ |
| **Unembedding:** | $2d_{\text{model}}n_{\text{vocab}}$ |
| **Total Transformer model** $\text{FLOPS}_{\text{Att,model}}$: | — |

### B.4 MEMORY OPERATION COUNTS

In this section, we count the memory operations for the mLSTM and the Transformer model architecture. We follow the same outline as for the FLOP counts in Appendix B.3 and first count the memory operations for the mLSTM cell (B.4.1) and the Self-Attention layer (B.4.3), and then combine them with the memory operations of the other layers in the model backbone to obtain the total memory operations for the mLSTM (B.4.2) and the Transformer model (B.4.4). We model weight MemOps as a one-time streaming cost (perfect on-chip reuse), i.e., independent of the number of token in the batch $BT$. This is reasonable with persistent/fused kernels and per-rank weight matrices that fit in on-chip cache. Depending on the exact experimental configuration, this assumption might not hold as we observe when modeling the step time through MemOps in Section C.3.

We include the memory operation count for the normalization layers, but can neglect them by setting $\text{bytes}_{\text{norm}} = 0$ and $\text{bytes}_{\text{act,norm}} = 0$.

#### B.4.1 MLSTM CELL MEMOPS

Similar to the FLOP counts in Appendix B.3.1, we count the memory operations for the mLSTM cell for both the chunkwise-parallel and the recurrent formulation.

**Chunkwise-Parallel Formulation (Tab. 13, Eq. 17).** The implementation of the chunkwise-parallel mLSTM formulation consists of two kernels (Beck et al., 2025a). We count the memory operations for the loading and storing of the inputs and outputs of each kernel for a single chunk and head in Tab. 13.

By multiplying with the number of heads $n_{\text{head}}$ and the number of chunks $n_{\text{chunk}} = T/L$, we obtain the total memory operation counts for the chunkwise-parallel mLSTM formulation as

$$\text{Bytes}_{\text{mLSTM,cwp}} = n_{\text{head}} \frac{T}{L} \left( 4L \times \text{bytes}_{\text{if}} + 3L \left( d_{\text{hv}} + d_{\text{qk}} \right) \times \text{bytes}_{\text{qkv}} \right. \tag{17}$$
$$\left. + 2n_{\text{head}} \left( L + d_{\text{hv}} d_{\text{qk}} + d_{\text{qk}} + 1 \right) \times \text{bytes}_{Cmn} \right).$$

Table 13: **Memory operation counts** for the **chunkwise-parallel mLSTM formulation**. All terms denote the memory operation count per head and chunk.

| Bytes | |
|---|---|
| *Inter-chunk Recurrent Kernel* | |
| **Load:** | $L(d_{\text{qk}} + d_{\text{hv}}) \times \text{bytes}_{\text{qkv}} + 2L \times \text{bytes}_{\text{if}}$ |
| **Store:** | $(d_{\text{qk}} d_{\text{hv}} + d_{\text{qk}} + 1) \times \text{bytes}_{Cnm}$ |
| *Intra-chunk Parallel Kernel* | |
| **Load:** | $L(2d_{\text{qk}} + d_{\text{hv}}) \times \text{bytes}_{\text{qkv}} + 2L \times \text{bytes}_{\text{if}}$ $+(d_{\text{qk}} d_{\text{hv}} + d_{\text{qk}} + 1) \times \text{bytes}_{Cnm}$ |
| **Store:** | $Ld_{\text{hv}} \times \text{bytes}_{\text{qkv}} + 2L \times \text{bytes}_{Cnm}$ |
| **Total:** | $4L \times \text{bytes}_{\text{if}}$ $+3L \left( d_{\text{hv}} + d_{\text{qk}} \right) \times \text{bytes}_{\text{qkv}}$ $+2 \left( L + d_{\text{hv}} d_{\text{qk}} + d_{\text{qk}} + 1 \right) \times \text{bytes}_{Cmn}$ |

**Recurrent Formulation (Tab. 14, Eq. 18).** During text generation we use the recurrent formulation, which loads the previous memory state and the current input and stores the output and the next memory state. We obtain the total memory operation counts for the recurrent mLSTM formulation by multiplying the counts in Tab. 14 with the number of heads $n_{\text{head}}$:

$$\text{Bytes}_{\text{mLSTM,rec}} = n_{\text{head}} \times \left( 2 \times \text{bytes}_{\text{if}} + 2(d_{\text{hv}} + d_{\text{qk}}) \times \text{bytes}_{\text{qkv}} + 2d_{\text{hv}} d_{\text{qk}} \times \text{bytes}_{Cmn} \right). \tag{18}$$

#### B.4.2 MLSTM MODEL MEMOPS

The memory operations of each layer of the backone (excluding the mLSTM cell) consist of the input and output activations as well as the parameters. The inputs and outputs depend on the number of tokens $BT$ in the batch, whereas the parameters are independent of the number of tokens.

Table 14: **Memory operation counts** for the **recurrent mLSTM formulation**. All terms denote the memory operation count for a single timestep per head. We assume the states are materialized at every timestep.

| Bytes | |
| --- | --- |
| **Load:** | $(2d_{\text{qk}} + d_{\text{hv}}) \times \text{bytes}_{\text{qkv}} + 2 \times \text{bytes}_{\text{if}}$ 
 $+(d_{\text{qk}}d_{\text{hv}} + d_{\text{qk}} + 1) \times \text{bytes}_{Cmn}$ |
| **Store:** | $d_{\text{hv}} \times \text{bytes}_{\text{qkv}} + (d_{\text{qk}}d_{\text{hv}} + d_{\text{qk}} + 1) \times \text{bytes}_{Cmn}$ |
| **Total:** | $2 \times \text{bytes}_{\text{if}} + 2(d_{\text{hv}} + d_{\text{qk}}) \times \text{bytes}_{\text{qkv}}$ 
 $+2d_{\text{hv}}d_{\text{qk}} \times \text{bytes}_{Cmn}$ |

The total memory operations for each layer are the sum of the memory operations for the input and output activations and the parameters and are given in Tab. 15. By default, we assume that all weights are stored in the same precision and use the same number of bytes $\text{bytes}_W$ for all weights.

Table 15: **Memory Operation counts** for the **mLSTM Model**.

| Memory Ops in bytes | Input & Output Activations | Weights |
| --- | --- | --- |
| **Embeddings:** | $BTn_{\text{vocab}}d_{\text{model}} \times \text{bytes}_{W_{\text{emb}}}$ | |
| *mLSTM (single layer)* | | |
| **PreNorm:** | $BTd_{\text{model}} \times \text{bytes}_{\text{act,norm}}$ | $d_{\text{model}} \times \text{bytes}_{W_{\text{norm}}}$ |
| **QKV:** | $BT(d_{\text{model}} + n_{\text{head}}(2d_{\text{qk}} + d_{\text{hv}})) \times \text{bytes}_{\text{qkv}}$ | $d_{\text{model}}n_{\text{head}}(2d_{\text{qk}} + d_{\text{hv}}) \times \text{bytes}_{W_{\text{qkv}}}$ |
| **Inpute & Forget Gates:** | $2BT(d_{\text{model}} + n_{\text{head}}) \times \text{bytes}_{\text{if}}$ | $(2d_{\text{model}}n_{\text{head}} + 2n_{\text{head}}) \times \text{bytes}_{W_{\text{if}}}$ |
| **mLSTM Cell:** | $\text{Bytes}_{\text{mLSTM}}$ | — |
| **Output Gate:** | $BT(d_{\text{model}} + n_{\text{head}}d_{\text{hv}}) \times \text{bytes}_{\text{act}}$ | $d_{\text{model}}n_{\text{head}}d_{\text{hv}} \times \text{bytes}_{W_{\text{o}}}$ |
| **Output Norm:** | $BTn_{\text{head}}d_{\text{hv}} \times \text{bytes}_{\text{act,norm}}$ | $n_{\text{head}}d_{\text{hv}} \times \text{bytes}_{W_{\text{norm}}}$ |
| **Output Projection:** | $BT(d_{\text{model}} + n_{\text{head}}d_{\text{hv}}) \times \text{bytes}_{\text{act}}$ | $d_{\text{model}}n_{\text{head}}d_{\text{hv}} \times \text{bytes}_{W_{\text{out}}}$ |
| *Total mLSTM layer* $\text{Bytes}_{\text{mLSTM,layer}}$: | — | |
| *Feedforward (single layer)* | | |
| **PreNorm:** | $BTd_{\text{model}} \times \text{bytes}_{\text{act,norm}}$ | $d_{\text{model}} \times \text{bytes}_{W_{\text{norm}}}$ |
| **MLPs:** | $3BT(d_{\text{model}} + d_{\text{ff}}) \times \text{bytes}_{\text{act,ff}}$ | $3d_{\text{model}}d_{\text{ff}}\text{bytes}_{W_{\text{ff}}}$ |
| *Total Feedforward* $\text{Bytes}_{\text{ff,layer}}$: | — | |
| **Output Norm:** | $BTd_{\text{model}} \times \text{bytes}_{\text{act,norm}}$ | $d_{\text{model}} \times \text{bytes}_{W_{\text{norm}}}$ |
| **Unembedding:** | $BT(d_{\text{model}} + n_{\text{vocab}}) \times \text{bytes}_{\text{act}}$ | $d_{\text{model}}n_{\text{vocab}} \times \text{bytes}_{W_{\text{emb}}}$ |
| **Total mLSTM model** $N_{\text{mLSTM}}$: | — | |

### B.4.3 SELF-ATTENTION MEMOPS

Similar to the FLOP counts in Appendix B.3.3, we count the memory operations for a single Self-Attention head during training or prefill and generation.

These two cases have very different memory operation counts, as during training and prefill we need to load the full sequence of tokens only once, whereas during autoregressive generation we have to load all previous tokens $T_{\text{p}} + t_{\text{g}}$ (i.e. the whole KV cace) for each generated token.

We consider FlashAttention implementations for the Self-Attention operation (Dao, 2024), where the Attention logits are not materialized in HBM. Therefore, we only count the memory operations for loading the query, key and value inputs and the output of Self-Attention in Tab. 16.

**Self-Attention in Training and Prefill (Eq. 19).** During training and prefill we need to load the full sequence of $T$ or $T_{\text{p}}$ tokens only once. The total memory operation counts are given by

$$\text{Bytes}_{\text{Att,train-pref}} = \big(T(d_{\text{qk}} + d_{\text{hv}})(n_{\text{head,q}} + n_{\text{head,kv}})\big) \times \text{bytes}_{\text{qkv}}. \tag{19}$$

**Self-Attention in Generation (Eq. 21).** Similar to the FLOP counts in Appendix B.3.3, also the memory operation counts for the Self-Attention layer during generation depend on the number of previous tokens $T = T_{\text{p}} + t_{\text{g}}$, where $T_{\text{p}}$ is the number of prefill tokens and $t_{\text{g}}$ is the number of generated tokens so far.

Table 16: **Memory operation counts** for **FlashAttention**. For training and prefill $T = S$, while for generation $S = 1$.

| Bytes | Generic |
|---|---|
| **Load:** | $\left(Sd_{\mathrm{qk}}n_{\mathrm{head,q}} + T(d_{\mathrm{qk}} + d_{\mathrm{hv}})n_{\mathrm{head,kv}}\right) \times \mathrm{bytes}_{\mathrm{qkv}}$ |
| **Store:** | $Sd_{\mathrm{hv}}n_{\mathrm{head,q}} \times \mathrm{bytes}_{\mathrm{qkv}}$ |
| **Total:** | $\left(S(d_{\mathrm{qk}} + d_{\mathrm{hv}})n_{\mathrm{head,q}} + T(d_{\mathrm{qk}} + d_{\mathrm{hv}})n_{\mathrm{head,kv}}\right) \times \mathrm{bytes}_{\mathrm{qkv}}$ |

The number of memory operations for the $t_{\mathrm{g}}$-th generated token is given by

$$\mathrm{Bytes}_{\mathrm{Att,gen\text{-}step}}(t_{\mathrm{g}}) = \left((d_{\mathrm{qk}} + d_{\mathrm{hv}})n_{\mathrm{head,q}} + (T_{\mathrm{p}} + t_{\mathrm{g}})(d_{\mathrm{qk}} + d_{\mathrm{hv}})n_{\mathrm{head,kv}}\right) \times \mathrm{bytes}_{\mathrm{qkv}} \tag{20}$$

Similar to equations (12)-(15), we can compute the total number of memory operations for $T_{\mathrm{g}}$ generated tokens by summing up the per-step memory operations

$$\mathrm{Bytes}_{\mathrm{Att,gen\text{-}seq}} = \mathrm{bytes}_{\mathrm{qkv}} \times \left( T_{\mathrm{g}}(d_{\mathrm{qk}} + d_{\mathrm{hv}})n_{\mathrm{head,q}} \right.$$
$$\left. + \left(T_{\mathrm{p}}T_{\mathrm{g}} + \frac{1}{2}T_{\mathrm{g}}(T_{\mathrm{g}} + 1)\right)(d_{\mathrm{qk}} + d_{\mathrm{hv}})n_{\mathrm{head,kv}} \right) \tag{21}$$

### B.4.4 TRANSFORMER MODEL MEMOPS

Similar to the mLSTM backbone in Appendix B.4.2, the number of memory operations for the Transformer backbone (excluding the Self-Attention layer) consist of the input and output activations as well as the parameters. The memory operations for input and output activations depend on the number of tokens $BT$ in the batch, whereas the parameters are independent of the number of tokens.

The total memory operations for each layer are the sum of the memory operations for the input and output activations and the parameters and are given in Tab. 17. By default, we assume that all weights are stored in the same precision and use the same number of bytes $\mathrm{bytes}_W$ for all weights.

Table 17: **Memory Operation counts** for the **Transformer Model**.

| Memory Ops in bytes | Input & Output Activations | Weights |
|---|---|---|
| **Embeddings:** | $BTn_{\mathrm{vocab}}d_{\mathrm{model}} \times \mathrm{bytes}_{W_{\mathrm{emb}}}$ | |
| *Attention (single layer)* | | |
| **PreNorm:** | $BTd_{\mathrm{model}} \times \mathrm{bytes}_{\mathrm{act,norm}}$ | $d_{\mathrm{model}} \times \mathrm{bytes}_{W_{\mathrm{norm}}}$ |
| **QKV:** | $BT\left(d_{\mathrm{model}} + n_{\mathrm{head}}(2d_{\mathrm{qk}} + d_{\mathrm{hv}})\right) \times \mathrm{bytes}_{\mathrm{qkv}}$ | $d_{\mathrm{model}}\left(d_{\mathrm{qk}}n_{\mathrm{head,q}} + (d_{\mathrm{qk}} + d_{\mathrm{hv}})n_{\mathrm{head,kv}}\right) \times \mathrm{bytes}_{W_{\mathrm{qkv}}}$ |
| **Attention:** | $\mathrm{Bytes}_{\mathrm{Att}}$ | — |
| **Output Projection:** | $BT(d_{\mathrm{model}} + n_{\mathrm{head,q}}d_{\mathrm{hv}}) \times \mathrm{bytes}_{\mathrm{act}}$ | $d_{\mathrm{model}}n_{\mathrm{head,q}}d_{\mathrm{hv}} \times \mathrm{bytes}_{W_{\mathrm{out}}}$ |
| *Total Attention layer* $\mathrm{Bytes}_{\mathrm{Att,layer}}$: | — | |
| *Feedforward (single layer)* | | |
| **PreNorm:** | $BTd_{\mathrm{model}} \times \mathrm{bytes}_{\mathrm{act,norm}}$ | $d_{\mathrm{model}} \times \mathrm{bytes}_{W_{\mathrm{norm}}}$ |
| **MLPs:** | $3BT(d_{\mathrm{model}} + d_{\mathrm{ff}}) \times \mathrm{bytes}_{\mathrm{act,ff}}$ | $3d_{\mathrm{model}}d_{\mathrm{ff}}\mathrm{bytes}_{W_{\mathrm{ff}}}$ |
| *Total Feedforward* $\mathrm{Bytes}_{\mathrm{ff,layer}}$: | — | |
| **Output Norm:** | $BTd_{\mathrm{model}} \times \mathrm{bytes}_{\mathrm{act,norm}}$ | $d_{\mathrm{model}} \times \mathrm{bytes}_{W_{\mathrm{norm}}}$ |
| **Unembedding:** | $BT(d_{\mathrm{model}} + n_{\mathrm{vocab}}) \times \mathrm{bytes}_{\mathrm{act}}$ | $d_{\mathrm{model}}n_{\mathrm{vocab}} \times \mathrm{bytes}_{W_{\mathrm{emb}}}$ |
| **Total Transformer model** $N_{\mathrm{Att}}$: | — | |

## C    MODELING INFERENCE CHARACTERISTICS

In this section, we create a model of the theoretical runtimes of operations in the xLSTM and Transformer model architectures to model their inference characteristics (TTFT and step time). This theoretical model is based on the FLOP and the memory operation counts in Appendix B.

This theoretical model of inference characteristics has two purposes: First, it allows to investigate the theoretical differences in maximal inference speed between xLSTM and Transformer architectures and explain the empirically observed behavior. Second, based on TTFT and step time measurements for specific architecture configurations, it allows to predict the theoretical inference speed for other (possibly larger) configurations and take this into account for selecting the optimal architecture configuration based on our scaling laws. This is important if there are certain requirements on maximal TTFTs or step times for a particular use-case. With this theoretical model, it is easily possible to determine model configurations which satisfy those conditions.

### C.1    BACKGROUND: THEORETICAL RUNTIME

In order to estimate the total theoretical runtime of workloads on GPUs or TPUs, we can break down the runtime into three components (Austin et al., 2025, Part 1):

- **Compute time** $\tau_{\text{FLOPs}}$: The time it takes to perform the FLOPs of the workload on the GPU(s).
- **Memory time** $\tau_{\text{mem}}$: The time for memory loads and stores from and to GPU memory during a workload.
- **Communication time** $\tau_{\text{comm}}$: The time for communicating or transferring data (e.g. intermediate results) between multiple GPUs taking part in a workload.

Given the number of floating point operations $\text{FLOPs}_{\text{algo}}$, the number of bytes $\text{Bytes}_{\text{mem,algo}}$ that must be loaded and stored, and the number of bytes $\text{Bytes}_{\text{comm,algo}}$ that must be communicated between GPUs, we can compute the individual runtimes as

$$\tau_{\text{FLOPs,algo}} = \frac{\text{FLOPs}_{\text{algo}}}{\alpha_{\text{acc}}}, \quad \tau_{\text{mem,algo}} = \frac{\text{Bytes}_{\text{mem,algo}}}{\beta_{\text{acc}}} \quad \text{and} \quad \tau_{\text{comm,algo}} = \frac{\text{Bytes}_{\text{comm,algo}}}{\gamma_{\text{Bytes}}}, \quad (22)$$

where $\alpha_{\text{acc}}$, $\beta_{\text{acc}}$ and $\gamma_{\text{Bytes}}$ are the accelerator specific compute speed in FLOPs/s, the accelerator memory bandwidth in Bytes/s and the accelerator communication bandwidth in Bytes/s, respectively.

For accelerator speed $\alpha_{\text{acc}}$, accelerator memory bandwidth $\beta_{\text{acc}}$, and accelerator communication bandwidth $\gamma_{\text{Bytes}}$, we use the hardware specifications of NVIDIA V100[3], A100[4], H100[5] and B200[6] GPUs, which we summarize in Tab. 18.

Table 18: **Hardware Accelerator Specification** for NVIDIA GPUs used in this analysis. Values without sparsity. If only the value with sparsity is known, we divide by 2.

| GPU | Year | bfloat16 [FLOPs/s] | Memory Bandwidth [Byte/s] | Arithmetic Intensity [FLOPs/byte] | Communication Bandwidth [Byte/s] |
|---|---|---|---|---|---|
| V100 SXM2 | 2017 | 120e12 | 0.9e12 | 133 | 0.3e12 |
| A100 SXM | 2020 | 312e12 | 2.039e12 | 161 | 0.6e12 |
| H100 SXM | 2022 | 989e12 | 3.35e12 | 295 | 0.9e12 |
| B200 HGX | 2025 | 2250e12 | 7.7e12 | 292 | 1.8e12 |

If there is no overlap between computation and memory or communication operations, or in other words if we cannot load, store or communicate data while the GPU is doing FLOPs, the total runtime

---

[3]https://www.nvidia.com/en-au/data-center/v100/
[4]https://www.nvidia.com/en-us/data-center/a100/
[5]https://www.nvidia.com/en-au/data-center/h100/
[6]https://resources.nvidia.com/en-us-blackwell-architecture/datasheet

is the sum of the two, i.e.

$$\tau_{\text{algo,upper}} = \tau_{\text{FLOPs,algo}} + \tau_{\text{mem/comm,algo}}. \quad (23)$$

If the computation and memory or communication operations can be overlapped (i.e. happen in parallel), the total runtime is the maximum of the two, i.e.

$$\tau_{\text{algo,lower}} = \max\left(\tau_{\text{FLOPs,algo}}, \tau_{\text{mem/comm,algo}}\right). \quad (24)$$

This means the runtime is lower bounded by the maximum of the two and upper bounded by their sum (Austin et al., 2025, Part 1).

**Roofline model.** A helpful model for determining whether runtime is bounded by computation (compute-bound) or by memory/bandwidth (memory-bound) is the roofline model (Williams et al., 2009), see Figure 13 for an illustration. The roofline relates the attainable FLOPs/s with the arithmetic intensity $I_{\text{algo}}$ of the operation performed on the GPU which is given by

$$I_{\text{algo}} = \frac{\text{FLOPs}_{\text{algo}}}{\text{Bytes}_{\text{algo}}}. \quad (25)$$

Thus, the arithmetic intensity is the FLOPs per byte for a given operation. When the arithmetic intensity of operations increases, the attainable FLOPs/s increase linearly - operations are essentially memory-bound; the GPU has to wait for bytes to arrive to perform calculations. In this setting, the runtime is effectively given by $\tau_{\text{mem/comm,algo}}$.

Upon reaching the arithmetic intensity of the accelerator $I_{\text{acc}}$ (see Tab. 18 for specifications for common GPU types), the "roofline" is reached and operations are essentially compute bound; the GPU still performs calculations while the next inputs are ready. In this setting, the runtime is effectively given by $\tau_{\text{FLOPs,algo}}$.

**Inference stages.** As outlined in Section 4, inference with LLMs is typically split into two stages, prefill and generation.

For the *prefill stage*, the TTFT is the key performance metric which is the runtime of the LLM in processing an input sequence if a certain prefill length, building up caches (Transformer) / memory cells (xLSTM) and generating the first token. Following

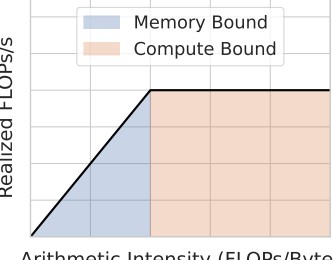

Figure 13: Roofline model.

Austin et al. (2025, Part 7), we assume that even at relatively low prefill lengths of 256, inference is dominated by large matrix multiplications for both Transformers and xLSTM and therefore consider the prefill stage the be compute bound. While this might not perfectly model very small prefill lengths, those are generally dominated by constant overheads.

For the *generation stage*, step time is the key performance metric which is the runtime of the LLM in generating a new token after having processed the the whole input sequence up to the last token. This means that during a forward pass, only a tiny amount of compute is necessary to account for this new token. However, for Transformers it is necessary to load from the KV cache, which is a very bandwidth-intensive operations, followed by streaming weights and storing and loading activations for both architectures. Consequently, arithmetic intensities during generation are generally rather low (see also Austin et al., 2025, Part 7). We thus assume that during the generation stage, both Transformers and xLSTM are memory bound.

### C.2 PREFILL STAGE: TIME TO FIRST TOKEN

As we assume to be compute bound during prefill, we model the runtime of the prefill stage which corresponds to the TTFT as (c.f. Eq. (4)):

$$\tau_{\text{FLOPs,algo}} = \frac{\text{FLOPs}_{\text{algo}}}{\alpha_{\text{eff}}} + \epsilon. \quad (26)$$

$\text{FLOPs}_{\text{algo}}$ can be calculated analytically given the FLOPs calculations provided in Appendix B.3, $\alpha_{\text{eff}}$ and $\epsilon$ need to be fitted using the measured data. Exemplarily, we show the runtimes fitted for the measured TTFT in Figure 14 (Transfomer) and Figure 15 (xLSTM) for different model sizes. We fit $\alpha_{\text{eff}}$ and $\epsilon$ per model configuration on TTFTs obtained under various combinations of batch sizes and prefill lengths. Our fits show excellent agreement between the predictions from our quantitative

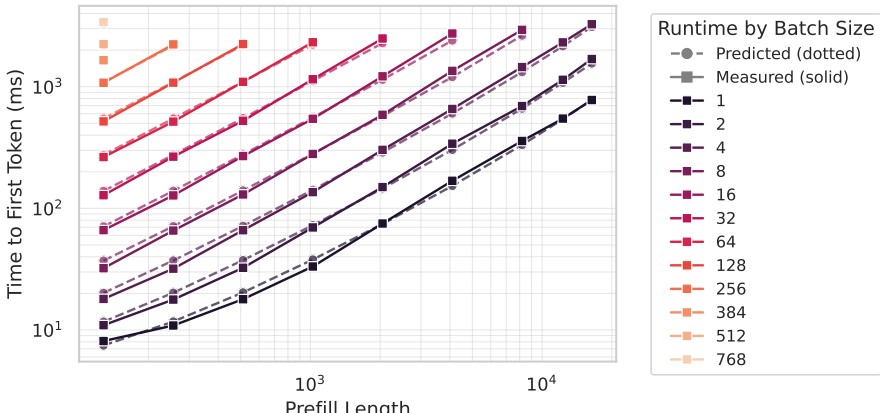

Figure 14: Time to first token, measured and fitted, for a 7B Transformer model as a function of prefill for different batch sizes.

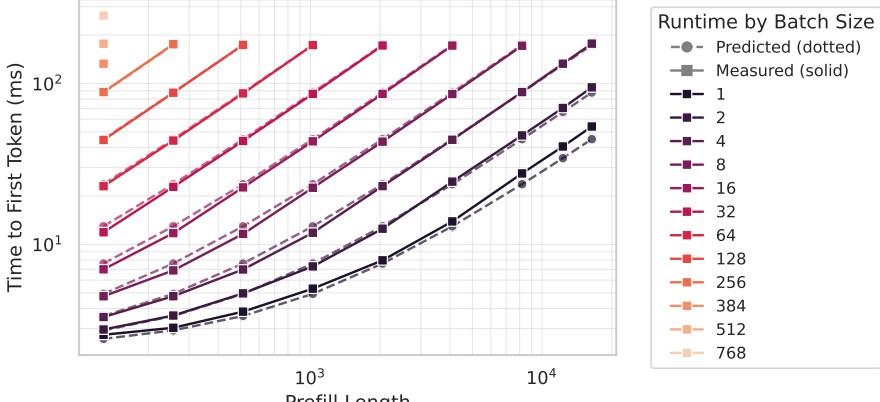

Figure 15: Time to first token, measured and fitted, for a 400M xLSTM model as a function of prefill for different batch sizes.

runtime model and the measured data. In Figure 16 we further show the quotient of the fitted $\alpha_{\mathrm{eff}}$ and the hardware parameter $\alpha_{\mathrm{acc}}$ for all model sizes. If $\alpha_{\mathrm{eff}}/\alpha_{\mathrm{acc}} = 1$, the hardware would be perfectly utilized according to our model. We see that for both Transformers and xLSTM, the quotient increases, thus larger models utilize the hardware better. Furthermore, both models show relatively similar trends and magnitudes, indicating that the empirical measurement setup allowed for a fair comparison.

## C.3 GENERATION STAGE: STEP TIME

As we assume to be memory-bound during generation stage, we model the runtime of the generation stage which corresponds to the step time as (c.f. Eq. 4):

$$\tau_{\mathrm{mem,algo}} = \frac{\mathrm{Bytes}_{\mathrm{mem,algo}}}{\beta_{\mathrm{eff}}} + \epsilon \,. \tag{27}$$

$\mathrm{Bytes}_{\mathrm{mem,algo}}$ can be calculated analytically given the MemOps calculations provided in Appendix B.4, $\beta_{\mathrm{eff}}$ and $\epsilon$ need to be fitted using the measured data. Furthermore, we found that the fit quality for Transformer further improved by fitting another constant that scales with the batch size. Exemplarily, we show the runtimes fitted for the measured step times in Figure 17 (Transformer) and Figure 18 (xLSTM) for different model sizes. Again, we find a very good agreement between the predictions from our quantitative runtime model and the measured data.

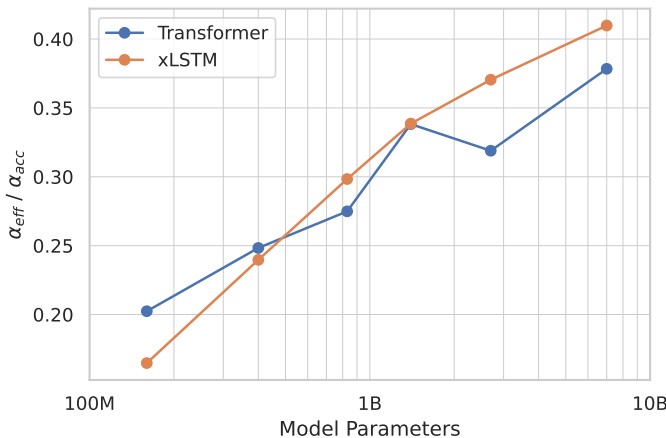

Figure 16: Comparing the fitted $\alpha_{\text{eff}}$ to the accelerator $\alpha_{\text{acc}}$ ($989e12$ for a H100 see Tab. 18). With our experimental setup, we attain similar effective FLOPs for both the Transformer and xLSTM. As expected, the accelerator is better utilized by larger models.

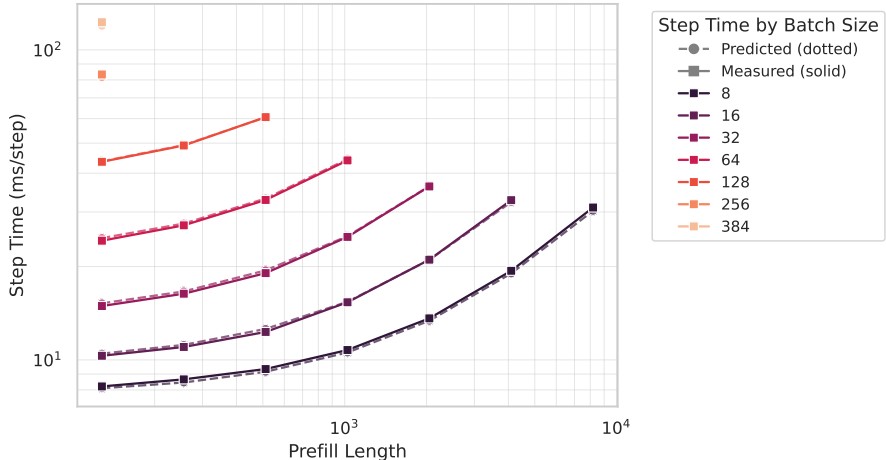

Figure 17: Step time, measured and fitted, for a 7B Transformer model as a function of prefill for different batch sizes.

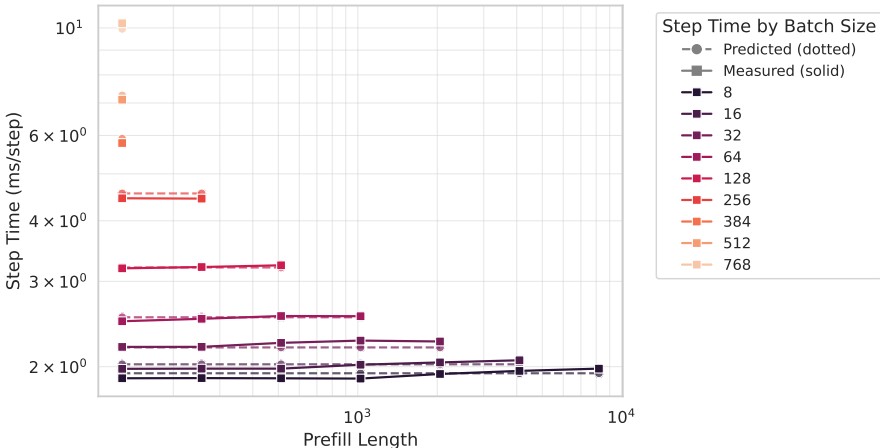

Figure 18: Step time, measured and fitted, for a 400M xLSTM model as a function of prefill for different batch sizes.

# D MODEL CONFIGURATIONS

In this section, we list the model hyperparameters and sizes of all training runs in Token/Param (Sec. D.1) and IsoFLOP (Sec. D.2) of the dataset for our scaling law study.

## D.1 MODEL SIZES AND HYPERPARAMETERS IN TOKEN/PARAM CONFIGURATION

Table 19: **List of hyperparameters for xLSTM models** trained with the **Token/Param** configuration with context length $T = 8192$.

| #Params (M) | $d_{\text{model}}$ | $d_{\text{ff}}$ | $d_{\text{qk}}$ | $d_{\text{hv}}$ | $n_{\text{heads}}$ | $n_{\text{layer}}$ | $B$ (seqs) | LR |
|---|---|---|---|---|---|---|---|---|
| 164 | 768 | 2112 | 64 | 128 | 6 | 12 | 128 | 3e-3 |
| 406 | 1024 | 2752 | 128 | 256 | 4 | 24 | 128 | 3e-3, 1e-3 |
| 841 | 1536 | 4160 | 192 | 384 | 4 | 24 | 256 | 1e-3, 8e-4 |
| 1420 | 2048 | 5504 | 256 | 512 | 4 | 24 | 256 | 8e-4, 7e-4 |
| 2780 | 2560 | 6848 | 256 | 512 | 5 | 32 | 512 | 7e-4 |
| 6865 | 4096 | 10944 | 256 | 512 | 8 | 32 | 256, 512 | 5e-4, 4e-4 |

Table 20: **List of hyperparameters for Transformer models** trained with the **Token/Param** configuration with context length $T = 8192$.

| #Params (M) | $d_{\text{model}}$ | $d_{\text{ff}}$ | $d_{\text{hv}}$ | $n_{\text{heads}}$ | $n_{\text{layer}}$ | $B$ (seqs) | LR |
|---|---|---|---|---|---|---|---|
| 162 | 768 | 2048 | 64 | 12 | 12 | 128 | 3e-3, 1e-3 |
| 406 | 1024 | 2752 | 64 | 16 | 24 | 128 | 3e-3, 1e-3 |
| 834 | 1536 | 4096 | 96 | 16 | 24 | 256 | 1e-3 |
| 1420 | 2048 | 5504 | 128 | 16 | 24 | 256 | 8e-4 |
| 2779 | 2560 | 6848 | 80 | 32 | 32 | 512 | 7e-4 |
| 6863 | 4096 | 10944 | 128 | 32 | 32 | 256, 512 | 5e-4 |

## D.2  MODEL SIZES AND HYPERPARAMETERS IN ISOFLOP CONFIGURATION

Table 21: **List of hyperparameters for xLSTM models** trained with the **IsoFLOP** configuration.

| #Params (M) | $d_{\text{model}}$ | $d_{\text{ff}}$ | $d_{\text{qk}}$ | $d_{\text{hv}}$ | $n_{\text{heads}}$ | $n_{\text{layer}}$ |
|---|---|---|---|---|---|---|
| 83 | 512 | 1408 | 64 | 128 | 4 | 10 |
| 90 | 512 | 1408 | 64 | 128 | 4 | 12 |
| 96 | 512 | 1408 | 64 | 128 | 4 | 14 |
| 102 | 512 | 1408 | 64 | 128 | 4 | 16 |
| 114 | 640 | 1728 | 64 | 128 | 5 | 10 |
| 123 | 640 | 1728 | 64 | 128 | 5 | 12 |
| 128 | 640 | 1728 | 64 | 128 | 5 | 13 |
| 133 | 640 | 1728 | 64 | 128 | 5 | 14 |
| 143 | 640 | 1728 | 64 | 128 | 5 | 16 |
| 164 | 768 | 2112 | 64 | 128 | 6 | 12 |
| 185 | 768 | 2112 | 64 | 128 | 6 | 15 |
| 207 | 896 | 2432 | 64 | 128 | 7 | 12 |
| 207 | 768 | 2112 | 64 | 128 | 6 | 18 |
| 236 | 896 | 2432 | 64 | 128 | 7 | 15 |
| 265 | 896 | 2432 | 64 | 128 | 7 | 18 |
| 295 | 896 | 2432 | 64 | 128 | 7 | 21 |
| 324 | 896 | 2432 | 64 | 128 | 7 | 24 |
| 330 | 1024 | 2752 | 128 | 256 | 4 | 18 |
| 353 | 896 | 2432 | 64 | 128 | 7 | 27 |
| 368 | 1024 | 2752 | 128 | 256 | 4 | 21 |
| 406 | 1024 | 2752 | 128 | 256 | 4 | 24 |
| 444 | 1024 | 2752 | 128 | 256 | 4 | 27 |
| 482 | 1024 | 2752 | 128 | 256 | 4 | 30 |
| 503 | 1152 | 3136 | 64 | 128 | 9 | 24 |
| 552 | 1152 | 3136 | 64 | 128 | 9 | 27 |
| 601 | 1152 | 3136 | 64 | 128 | 9 | 30 |
| 604 | 1280 | 3456 | 128 | 256 | 5 | 24 |
| 664 | 1280 | 3456 | 128 | 256 | 5 | 27 |
| 715 | 1408 | 3776 | 64 | 128 | 11 | 24 |
| 724 | 1280 | 3456 | 128 | 256 | 5 | 30 |
| 787 | 1408 | 3776 | 64 | 128 | 11 | 27 |
| 841 | 1536 | 4160 | 128 | 256 | 6 | 24 |
| 859 | 1408 | 3776 | 64 | 128 | 11 | 30 |
| 927 | 1536 | 4160 | 128 | 256 | 6 | 27 |
| 1013 | 1536 | 4160 | 128 | 256 | 6 | 30 |
| 1108 | 1792 | 4800 | 128 | 256 | 7 | 24 |
| 1224 | 1792 | 4800 | 128 | 256 | 7 | 27 |
| 1340 | 1792 | 4800 | 128 | 256 | 7 | 30 |
| 1421 | 2048 | 5504 | 128 | 256 | 8 | 24 |
| 1573 | 2048 | 5504 | 128 | 256 | 8 | 27 |
| 1772 | 2304 | 6208 | 128 | 256 | 9 | 24 |
| 1876 | 2048 | 5504 | 128 | 256 | 8 | 33 |
| 1964 | 2304 | 6208 | 128 | 256 | 9 | 27 |
| 2028 | 2048 | 5504 | 128 | 256 | 8 | 36 |
| 2157 | 2304 | 6208 | 128 | 256 | 9 | 30 |
| 2350 | 2304 | 6208 | 128 | 256 | 9 | 33 |
| 2781 | 2560 | 6848 | 128 | 256 | 10 | 32 |
| 3017 | 2560 | 6848 | 128 | 256 | 10 | 35 |
| 3150 | 2816 | 7552 | 128 | 256 | 11 | 30 |
| 3254 | 2560 | 6848 | 128 | 256 | 10 | 38 |
| 3342 | 2816 | 7552 | 128 | 256 | 11 | 32 |
| 3533 | 2816 | 7552 | 128 | 256 | 11 | 34 |
| 3724 | 2816 | 7552 | 128 | 256 | 11 | 36 |
| 3726 | 3072 | 8256 | 128 | 256 | 12 | 30 |
| 3954 | 3072 | 8256 | 128 | 256 | 12 | 32 |
| 4410 | 3072 | 8256 | 128 | 256 | 12 | 36 |
| 4597 | 3328 | 8896 | 128 | 256 | 13 | 32 |
| 5130 | 3328 | 8896 | 128 | 256 | 13 | 36 |
| 5311 | 3584 | 9600 | 128 | 256 | 14 | 32 |
| 5930 | 3584 | 9600 | 128 | 256 | 14 | 36 |
| 6464 | 4096 | 10944 | 128 | 256 | 16 | 30 |
| 6867 | 4096 | 10944 | 128 | 256 | 16 | 32 |

Table 22: **List of hyperparameters for Transformer models** trained with the **IsoFLOP** configuration.

| #Params (M) | $d_{\text{model}}$ | $d_{\text{ff}}$ | $d_{\text{v}}$ | $n_{\text{heads}}$ | $n_{\text{layer}}$ |
|---|---|---|---|---|---|
| 83 | 512 | 1408 | 64 | 8 | 10 |
| 90 | 512 | 1408 | 64 | 8 | 12 |
| 96 | 512 | 1408 | 64 | 8 | 14 |
| 102 | 512 | 1408 | 64 | 8 | 16 |
| 113 | 640 | 1728 | 64 | 10 | 10 |
| 128 | 640 | 1728 | 64 | 10 | 13 |
| 133 | 640 | 1728 | 64 | 10 | 14 |
| 143 | 640 | 1728 | 64 | 10 | 16 |
| 162 | 768 | 2048 | 64 | 12 | 12 |
| 183 | 768 | 2048 | 64 | 12 | 15 |
| 204 | 768 | 2048 | 64 | 12 | 18 |
| 207 | 896 | 2432 | 64 | 14 | 12 |
| 236 | 896 | 2432 | 64 | 14 | 15 |
| 265 | 896 | 2432 | 64 | 14 | 18 |
| 294 | 896 | 2432 | 64 | 14 | 21 |
| 324 | 896 | 2432 | 64 | 14 | 24 |
| 330 | 1024 | 2752 | 64 | 16 | 18 |
| 368 | 1024 | 2752 | 64 | 16 | 21 |
| 406 | 1024 | 2752 | 64 | 16 | 24 |
| 444 | 1024 | 2752 | 64 | 16 | 27 |
| 482 | 1024 | 2752 | 64 | 16 | 30 |
| 498 | 1152 | 3072 | 128 | 9 | 24 |
| 545 | 1152 | 3072 | 128 | 9 | 27 |
| 593 | 1152 | 3072 | 128 | 9 | 30 |
| 604 | 1280 | 3456 | 128 | 10 | 24 |
| 664 | 1280 | 3456 | 128 | 10 | 27 |
| 714 | 1408 | 3776 | 128 | 11 | 24 |
| 723 | 1280 | 3456 | 128 | 10 | 30 |
| 786 | 1408 | 3776 | 128 | 11 | 27 |
| 834 | 1536 | 4096 | 128 | 12 | 24 |
| 858 | 1408 | 3776 | 128 | 11 | 30 |
| 919 | 1536 | 4096 | 128 | 12 | 27 |
| 1003 | 1536 | 4096 | 128 | 12 | 30 |
| 1107 | 1792 | 4800 | 128 | 14 | 24 |
| 1223 | 1792 | 4800 | 128 | 14 | 27 |
| 1339 | 1792 | 4800 | 128 | 14 | 30 |
| 1420 | 2048 | 5504 | 128 | 16 | 24 |
| 1572 | 2048 | 5504 | 128 | 16 | 27 |
| 1723 | 2048 | 5504 | 128 | 16 | 30 |
| 1760 | 2304 | 6144 | 128 | 18 | 24 |
| 1951 | 2304 | 6144 | 128 | 18 | 27 |
| 2142 | 2304 | 6144 | 128 | 18 | 30 |
| 2334 | 2304 | 6144 | 128 | 18 | 33 |

# E   COMPUTE OPTIMAL PARAMETER, TOKEN AND FLOP COUNT ESTIMATES

In this section, we determine compute-optimal training setups for various model sizes based on the scaling laws derived from our IsoFLOP approach in sections 3.4 and 3.5. In Section E.1, we show the configurations for our power laws obtained for a context length of 8192 (see Figures 4 and 9), while in Section E.2 we present the compute optimal configurations obtained from our power law fits for varying context lengths (see Figures 5 and 10). The power law fits for Section E.2 contain fewer IsoFLOP profiles than the fits for Section E.1.

We construct these tables by first choosing a range of model sizes, then identifying the optimal compute budget associated with each size (for example, from Figures 4 or 5), and finally inferring the corresponding optimal number of training tokens, such as from Figures 9 or 10.

Across all tables in sections E.1 and E.2, we observe that Transformer models have a higher compute-optimal token-to-parameter ratio than xLSTM models.

Moreover, in contrast to the Chinchilla scaling laws, which find that the optimal token-to-parameter ratio is constant at around 22 across model sizes (Hoffmann et al., 2022, Table 3), our compute optimal token-to-parameter ratio decreases for larger models. This difference arises primarily from the distinct exponents in the scaling laws (Ours: $a = 0.575$, $b = 0.424$ vs. (Hoffmann et al., 2022, Table 2): $a = 0.49(0.462, 0.534)$, $b = 0.51(0.483, 529)$). Porian et al. (2024) have investigated these discrepancies and found the root cause to be in the learning rate decay for the training runs in the IsoFLOP configurations (see also Appendix A.3). They found exponents comparable to those in our work and were able to reproduce the Chinchilla scaling law exponents by using a fixed learning rate across all IsoFLOP training runs.

## E.1   COMPUTE OPTIMAL CONFIGURATIONS FOR CONTEXT LENGTH 8192

Table 23: Estimated optimal training FLOPs, Tokens, and Token/Param Ratio for varying model sizes from IsoFLOP power-law fits for **Transformer and xLSTM models** trained with context length 8192. The table is obtained from Figures 4 and 9.

| #Params | Transformer | | | xLSTM | | |
|---|---|---|---|---|---|---|
| | #FLOPs $A' = 0.0023$ $a = 0.575$ | #Tokens $B' = 58.5$ $b = 0.424$ | Token/ Param Ratio | #FLOPs $A' = 0.012$ $a = 0.547$ | #Tokens $B' = 77.7$ $b = 0.417$ | Token/ Param Ratio |
| 100M | 3.24e18 | 4.17B | 41.7 | 1.33e18 | 2.83B | 28.3 |
| 400M | 3.61e19 | 11.6B | 29.0 | 1.68e19 | 8.15B | 20.4 |
| 1B | 1.78e20 | 22.8B | 22.8 | 8.97e19 | 16.4B | 16.4 |
| 2B | 5.94e20 | 38.1B | 19.0 | 3.18e20 | 27.8B | 13.9 |
| 4B | 1.98e21 | 63.5B | 15.9 | 1.13e21 | 47.1B | 11.8 |
| 8B | 6.62e21 | 106B | 13.2 | 4.01e21 | 79.9B | 10.0 |
| 10B | 9.76e21 | 125B | 12.5 | 6.03e21 | 94.8B | 9.5 |
| 14B | 1.75e22 | 160B | 11.4 | 1.11e22 | 122B | 8.7 |
| 32B | 7.38e22 | 295B | 9.2 | 5.05e22 | 230B | 7.2 |
| 67B | 2.67e23 | 508B | 7.6 | 1.95e23 | 404B | 6.0 |
| 175B | 1.42e24 | 1.03T | 5.9 | 1.13e24 | 840B | 4.8 |

## E.2 COMPUTE OPTIMAL CONFIGURATIONS FOR VARYING CONTEXT LENGTHS

Table 24: Estimated optimal training FLOPs, Tokens, and Token/Param Ratio across context lengths from IsoFLOP context-specific power-law fits for **Transformer models**. The table is obtained from Figures 5 and 10.

| ctx length: | 2048 | | | 8192 | | | 16384 | | |
|---|---|---|---|---|---|---|---|---|---|
| | #FLOPs $A' = 0.0069$ $a = 0.553$ | #Tokens $B' = 74$ $b = 0.423$ | Token/ Param Ratio | #FLOPs $A' = 0.0021$ $a = 0.577$ | #Tokens $B' = 65$ $b = 0.422$ | Token/ Param Ratio | #FLOPs $A' = 0.0025$ $a = 0.569$ | #Tokens $B' = 34.5$ $b = 0.432$ | Token/ Param Ratio |
| #Params | | | | | | | | | |
| 100M | 2.27e18 | 4.24B | 42.4 | 3.26e18 | 4.19B | 41.9 | 4.19e18 | 3.91B | 39.1 |
| 400M | 2.78e19 | 12.2B | 30.5 | 3.59e19 | 11.5B | 28.8 | 4.78e19 | 11.2B | 28.0 |
| 1B | 1.46e20 | 24.6B | 24.6 | 1.76e20 | 22.5B | 22.5 | 2.39e20 | 22.5B | 22.5 |
| 2B | 5.09e20 | 41.7B | 20.8 | 5.84e20 | 37.4B | 18.7 | 8.07e20 | 38.1B | 19.0 |
| 4B | 1.78e21 | 70.8B | 17.7 | 1.94e21 | 62.1B | 15.5 | 2.73e21 | 64.5B | 16.1 |
| 8B | 6.23e21 | 120B | 15.0 | 6.44e21 | 103B | 12.9 | 9.21e21 | 109B | 13.6 |
| 10B | 9.32e21 | 143B | 14.3 | 9.48e21 | 121B | 12.1 | 1.36e22 | 129B | 12.9 |
| 14B | 1.71e22 | 184B | 13.1 | 1.7e22 | 155B | 11.1 | 2.46e22 | 167B | 11.9 |
| 32B | 7.62e22 | 346B | 10.8 | 7.11e22 | 284B | 8.9 | 1.05e23 | 313B | 9.8 |
| 67B | 2.9e23 | 609B | 9.1 | 2.56e23 | 487B | 7.3 | 3.85e23 | 549B | 8.2 |
| 175B | 1.64e24 | 1.27T | 7.3 | 1.35e24 | 982B | 5.6 | 2.08e24 | 1.14T | 6.5 |

Table 25: Estimated optimal training FLOPs, Tokens, and Token/Param Ratio across context lengths from IsoFLOP context-specific power-law fits for **xLSTM models**. The table is obtained from Figures 5 and 10.

| ctx length: | 2048 | | | 8192 | | | 16384 | | |
|---|---|---|---|---|---|---|---|---|---|
| | #FLOPs $A' = 0.0086$ $a = 0.555$ | #Tokens $B' = 141$ $b = 0.403$ | Token/ Param Ratio | #FLOPs $A' = 0.0161$ $a = 0.541$ | #Tokens $B' = 46.8$ $b = 0.429$ | Token/ Param Ratio | #FLOPs $A' = 0.005$ $a = 0.566$ | #Tokens $B' = 336$ $b = 0.385$ | Token/ Param Ratio |
| #Params | | | | | | | | | |
| 100M | 1.32e18 | 2.83B | 28.3 | 1.3e18 | 2.74B | 27.4 | 1.58e18 | 3.44B | 34.4 |
| 400M | 1.6e19 | 7.73B | 19.3 | 1.69e19 | 8.22B | 20.6 | 1.83e19 | 8.85B | 22.1 |
| 1B | 8.32e19 | 15B | 15.0 | 9.21e19 | 17B | 17.0 | 9.23e19 | 16.5B | 16.5 |
| 2B | 2.9e20 | 24.9B | 12.4 | 3.32e20 | 29.5B | 14.8 | 3.14e20 | 26.5B | 13.2 |
| 4B | 1.01e21 | 41.1B | 10.3 | 1.2e21 | 51B | 12.8 | 1.07e21 | 42.4B | 10.6 |
| 8B | 3.51e21 | 68B | 8.5 | 4.31e21 | 88.4B | 11.0 | 3.64e21 | 68B | 8.5 |
| 10B | 5.25e21 | 79.9B | 8.0 | 6.52e21 | 106B | 10.6 | 5.39e21 | 79.1B | 7.9 |
| 14B | 9.62e21 | 102B | 7.3 | 1.21e22 | 138B | 9.9 | 9.77e21 | 99.5B | 7.1 |
| 32B | 4.26e22 | 186B | 5.8 | 5.6e22 | 266B | 8.3 | 4.21e22 | 175B | 5.5 |
| 67B | 1.61e23 | 318B | 4.7 | 2.2e23 | 477B | 7.1 | 1.55e23 | 289B | 4.3 |
| 175B | 9.07e23 | 638B | 3.6 | 1.3e24 | 1.02T | 5.8 | 8.47e23 | 555B | 3.2 |

