# OpenReview forum: "xLSTM Scaling Laws: Competitive Performance with Linear Time-Complexity"
_ICLR.cc/2026/Conference — ICLR 2026 Poster_

### Official Review · Reviewer_H1t8 · 2025-10-16

**Soundness:** 3
**Presentation:** 4
**Contribution:** 3
**Rating:** 4
**Confidence:** 3

**Summary:**

This paper points out that the authors systematically compared the scaling laws of the performance-optimized xLSTM architecture with those of the dense multi-head self-attention Transformer architecture. The authors conducted comparative analyses from perspectives such as Training, Context length, and Inference.

**Strengths:**

1. This paper is well-written, with clear logic and concise readability.

2. It conducts extensive experiments, including comparative experiments on models of different types and scales.

3. The authors provide a wide range of comprehensive evaluation metrics.

**Weaknesses:**

1. First, this paper lacks sufficient novelty. Models based on the xLSTM architecture have already demonstrated significant advantages in areas such as inference in previous studies. However, this paper only conducts extended comparative experiments based on this existing foundation, which casts doubt on its innovativeness.

2. Second, compared to models with quadratic complexity, the design of LLM architectures based on linear complexity is inherently intended to reduce computational costs and achieve more significant benefits during training and inference—and this is a well-known fact. The paper conducts comparative analyses from perspectives including Training, Context length, and Inference, but these experiments are carried out under the premise that such performance characteristics are already known, resulting in insufficient contributions from the paper.

3. Finally, to fully compare the performance of models with linear complexity and quadratic complexity, limiting the scope solely to the xLSTM architecture is flawed. It is necessary to include a broader range of models and conduct more comprehensive result comparisons.

**Questions:**

See Weaknesses.

---

> ### Author Response · Authors · 2025-11-14
>
> We thank the reviewer H1t8 for their assessment of our work. We appreciate that the reviewer finds our paper well-written, our experiments extensive and our set of evaluation metrics comprehensive.
>
> We address the remaining weaknesses below:
>
> **Weaknesses:**
>
> Re 1 and 2):
> With this work we aim to investigate scaling laws of two existing language model architectures in training and the inference time scaling with model and context sizes.
>
> For the training scaling behavior, we are not aware of any other works showing that xLSTMs (or other Linear RNNs) maintain consistent power-law exponents in the overtraining regime, find that compute-optimal xLSTM models (or other Linear RNNs) are larger than their Transformer counterpart, or that the compute-optimal model size of xLSTMs (and thus also potentially other Linear RNNs) is robust to variations in context length.
>
>
> Previous work such as xLSTM 7B, conducted inference time experiments with the introduced xLSTM 7B model, but did not investigate other model sizes. In this work, we conduct extensive inference benchmarks across multiple model scales and investigate inference characteristics with theoretical runtime models based on FLOP and Memory Operation calculations that we fit from our benchmarks (See Section 4.2 and Appendix D).
>
> If you could point us to references or other related work we could have missed to cite or discuss, we are happy to add a discussion on them in the updated version of our paper.
>
> Re 3):
> Comparative studies at smaller scales to more models have been done in previous works. However, these studies often only varied one dimension – model size, dataset size (xLSTM) or scaled both with the same ratio (Mamba 1/2). Systematically established scaling laws for such architectures through the protocol of the Chinchilla paper were an open gap in the literature. Furthermore, little attention has been given so far to comparing scaling laws with different context lengths, which we investigate in this study.
>
> Instead of broadly comparing many model families, we deliberately focused our compute budget on an in-depth, large-scale comparison between one instance of the family of Linear RNN models – the xLSTM – to Transformer models in terms of compute budget, number of model parameters and number of training tokens, controlling either the compute budget in terms of FLOP or varying the token per parameter ratio.
> This enables insights on compute optimal model & dataset sizes, proves consistent scaling in the inference efficient overtraining regime and allows for comparisons under different context lengths.
> We are confident that we correctly state the scope and intent of our work, i.e. through the very specific title and clear definition of scope in abstract and introduction.
>
> Furthermore, we would like to point out that the results presented in this paper are already pushing the limits of academic compute budgets. For reference, the data point at 10^23 FLOPs for xLSTM (top right corner in Figure 2 right subplot) was obtained by training on 32 H100 Nodes for ~2 weeks.  Nevertheless, we see comparisons at even larger scales as an important direction for future work.
>
> We thank the reviewer for the feedback and hope they find their concerns well addressed.

---

### Official Review · Reviewer_dz4g · 2025-10-28

**Soundness:** 3
**Presentation:** 4
**Contribution:** 3
**Rating:** 8
**Confidence:** 3

**Summary:**

The authors study scaling behavior of a recently proposed sequence modeling architecture, finding it does well on many desirable axes compared to Transformer baselines.

**Strengths:**

- Study relevant regimes, incl. overtraining/inference time compute which are rarely paid attention to in scaling work even though they are principle considerations when training/serving modern frontier models.
- Science of scaling is cleanly and thoroughly done. Reproduction of important past work (power-law exponents from Chinchilla) is reassuring as a sanity check.
- Lots of large-scale empirics and careful parametric fits, very valuable contribution.

**Weaknesses:**

- Transformer baseline is weak and out of date (Llama-2 architecture in late 2025). There are a number of Transformer++ improvements that have appeared since Llama-2, all else fixed, that make big improvements to performance at scale (eg. RoPE, GQA, addressing attention sinks, no biases on linear layers, etc etc).
- I'm not convinced xLSTM actually outscale Transformers, and in fact would be willing to bet they do worse at frontier compute regimes. We can see the advantage decreasing with compute in Fig1(left), and we can see the faint Transformer curves overtaking the xLSTM ones on Fig1(right). This looks exactly like Fig4 of the Mamba [1] paper where Mamba was touted as "scaling favorably" but the gap shrunk slightly with compute in their plots. When things were scaled up, of course Transformers did much better than Mamba. I expect a similar thing to hold here (notice how the plot is missing Transformer at 1e23 where xLSTM begins to plateau, I would be very interested in seeing that last Transformer data point). Nonetheless the thorough and expansive empirics here are valuable, and this is how progress in architecture design is made, so this is a valuable paper.

**Questions:**

See weaknesses.

---

> ### Author Response · Authors · 2025-11-14
>
> We thank the reviewer dz4g for their assessment of our paper. We are pleased that our study design was found to be aligned with relevant questions for training/serving frontier models, conducted cleanly and thoroughly, and featuring expansive empirical results. Regarding the stated weaknesses:
>
> **Weaknesses:**
>
> Re 1):
> Our transformer models use RoPE as positional encoding, and we also omit biases in linear layers. We agree there are many other new architecture innovations, such as hybrid architectures, sliding window attention (in which case we would have to account for sink tokens) and investigating their effect on scaling laws is an important direction.
>
> With this work we aim to make a step in this direction, by rigorously comparing scaling laws in compute optimal and overtraining regimes of two language model architectures (xLSTM & Transformer) with different compute complexity w.r.t the context length. To enable sanity checks and cross comparisons to other scaling law papers, we had to use the same Transformer architecture as in the earlier papers.
>
> Re 2):
> We agree, doing architecture research at even larger scales is an important direction for future research. In this paper, we tried to do the largest possible study within our budgets and provide our insights in this paper. Our data point at 10^23 for xLSTM was obtained by training on 32 H100 Nodes for ~2 weeks. For our transformer baseline it would have been even longer – too long for our compute budget for this study. Also, we would like to emphasize that we do not necessarily see a plateau in the bottom right, there is likely a larger xLSTM trained on less tokens, which would achieve lower loss,, though this would need to be empirically confirmed. The last data point at 10^23 was not trained with the compute-optimal token/param ratio, but in the overtraining regime.
>
> Regarding Transformers overtaking xLSTM in Figure 1 (right), we agree that this happens at a certain number of tokens per parameter, i.e. individual fit lines per model size cross eventually. However, for the compute budget where Transformers would finally overtake, we find that there is a larger xLSTM trained on fewer tokens (thus with the same compute budget) that achieves lower loss.
>
> We thank the reviewer again for their invested time and feedback. We hope we could address all remaining open questions and concerns.

---

> > ### Comment · Reviewer_dz4g · 2025-11-27
> > **Official Comment by Reviewer**
> >
> > Thanks to the authors for their response. I will keep my score, though I will note upon rereading the paper and response my real opinion is that the paper lies somewhere between a 6 and an 8, if I had the option to give such a score I would.

---

### Official Review · Reviewer_LrJB · 2025-10-29

**Soundness:** 3
**Presentation:** 3
**Contribution:** 3
**Rating:** 6
**Confidence:** 3

**Summary:**

This paper presents a systematic empirical comparison of scaling behaviors between xLSTM and Transformer architectures for large language models. The study spans three dimensions: (1) training efficiency under compute-optimal and over-training regimes across 80M-7B parameters and 2B-2T tokens, (2) the relationship between optimal model size and context length, and (3) inference-time characteristics (TTFT and step time). The authors claim xLSTM is Pareto-dominant in training loss vs. compute, requires larger compute-optimal models, and shows widening advantages with increasing context length. This is a technically solid, large-scale empirical study that provides valuable evidence for xLSTM's scaling properties. The methodological innovations (accurate FLOP accounting, context-length scaling analysis) are notable contributions.

**Strengths:**

1. Comprehensive Experimental Scale：The study demonstrates an impressive experimental scale, encompassing 672 training runs with a total compute of 3.2×10²³ FLOPs. It systematically explores both the compute-optimal and over-training regimes—a distinction of significant practical relevance—while maintaining a fair and consistent comparison framework through unified training recipes, shared data (DCLM-BASELINE), and identical hyperparameter schedules.

2. Clear Empirical Findings: The paper presents clear and consistent empirical findings. xLSTM demonstrates Pareto dominance over Transformers, achieving lower loss at fixed compute across five orders of magnitude. It also exhibits remarkable over-training stability, with constant power-law exponents observed up to M=2200, confirming its reliability for inference-optimized training. Furthermore, xLSTM maintains strong context length robustness, as its optimal model size remains stable even when context length increases, whereas the Transformer's performance degrades notably.

**Weaknesses:**

1. **No Downstream Task Evaluation**. The entire paper focuses exclusively on pretraining cross-entropy loss without any downstream benchmarks (e.g., MMLU, HumanEval, common sense reasoning, summarization).

2. The context length experiments appear confounded, as different context lengths are trained on non-identical data distributions without clarification on how this issue is handled—for instance, whether through re-chunking, padding, or different data splits. Moreover, the y-axes in Figure 5 are not directly comparable across context lengths, and it remains unclear how the “tokens D” are counted when context size varies (i.e., whether they represent effective or nominal tokens). Clarifying the data preprocessing process or adding a fixed-dataset ablation where only context packing changes would strengthen the validity of these results.

**Questions:**

1. L024: Quantify "advantage widens", what is the relative improvement at 2K vs. 16K contexts?
3. L313: If losses aren't comparable across contexts, how should we interpret the intersecting curves in Figure 5 (right)?
4. L399: "Largest xLSTM has lower step time than smallest Transformer", is this specific to 16K prefill, or does it hold at shorter contexts (e.g., 512)?
5. It would be better to include at least one long-context benchmark evaluation (e.g., LongBench) to more convincingly demonstrate xLSTM’s effectiveness in handling extended context scenarios.

---

> ### Author Response · Authors · 2025-11-14
>
> We thank the reviewer LrJB for the feedback, which helps to improve our paper. We are pleased that they commend the comprehensive empirical scale and clear empirical findings of our work.
>
> **Weaknesses:**
>
> Re 1):
> Our study primarily focuses on comparing architectural performance, for which we employ loss and inference metrics as evaluation criteria. Specifically, we investigate scaling laws with respect to cross-entropy loss, consistent with prior work that has analyzed training and validation losses for similar purposes. Previous studies have demonstrated that validation loss serves as a reliable proxy for downstream performance, and our work builds upon this established relationship [1,2]. We explicitly acknowledge the absence of downstream task evaluations as a limitation, as discussed in the manuscript.
>
> Re 2):
> We thank the reviewer for raising this point.
> We will add further details on our data processing pipeline to the paper and clarify below, why the different context lengths are not confounding with the main message of the experiment in Figure 5.
>
> First, during all our experiments we use sequence packing (from https://google-grain.readthedocs.io/en/stable/_autosummary/grain.experimental.FirstFitPackIterDataset.html#grain-experimental-firstfitpackiterdataset), i.e. fill the batch with as many full documents as possible and add padding to the specific target sequence length. For our experiments in Figure 5, we set the target sequence length to 2k, 8k and 16k, respectively. We then count the nominal tokens, e.g. global batch size x sequence length x training steps (including the padding tokens).
>
> Second, to clarify our message in Figure 5:
> Our focus in Figure 5 is not a direct comparison between the validation losses between the three leftmost plots. Instead, we conduct independent FLOP controlled scaling law experiments at different context lengths and fit scaling laws for each context length.
> The three leftmost plots show the data that we use to produce these fits and the rightmost plot shows the fits. We can see a trend that validation losses for Transformers grow, because at constant compute budget they are trained on less tokens for longer context lengths due to more FLOPs being consumed by quadratic attention. For xLSTM the loss remains relatively constant, due to linear FLOP scaling in context length the total number of tokens for the same compute budget remains constant across context lengths. However, since these losses are not directly comparable due to sequence packing, we do not make claims about the relative differences.
>
> Instead, we make our main statements about the fits (which show optimal number of model parameters over compute, which is comparable across context sizes): Compute-optimal xLSTM models are larger than their Transformer counterparts and the compute-optimal model size of xLSTMs is robust to variations in context length.
>
> We will update the corresponding passages and add details about our data processing pipeline.
>
> [1] Gadre, S.Y., Smyrnis, G., Shankar, V., Gururangan, S., Wortsman, M., Shao, R., Mercat, J., Fang, A., Li, J., Keh, S. and Xin, R., 2024. Language models scale reliably with over-training and on downstream tasks. arXiv preprint arXiv:2403.08540.
>
> [2] Sardana, N., Portes, J., Doubov, S. and Frankle, J., 2023. Beyond chinchilla-optimal: Accounting for inference in language model scaling laws. arXiv preprint arXiv:2401.00448.

---

> > ### Author Response · Authors · 2025-11-14
> >
> > **Questions:**
> >
> > Re 1, L024):
> > This last sentence of the abstract was intended to summarize findings on context length in training and inference.
> > In inference we see that for larger prefill context lengths the step time and time to first tokens difference between Transformer and xLSTM grows (“widens”). E.g. compare the 2nd points from the left with the 3rd points from the right in Figure 6 (right).
> > For training (e.g. Figure 5) we cannot quantify it fully, since the validation losses are not directly comparable as we state in the paper and you pointed out correctly. We merely aimed to describe the general trend as described above (see Re 2) to weaknesses).
> > Thanks for pointing this out, we will update the last sentence in the abstract in the revision of our paper.
> >
> > Re 2, L313):
> > Note that the 2k and 16k curves of the mLSTM in the rightmost plot in Fig 5 are parallel, while only the 8k line seems to have a slightly different slope. We believe this is due to noise in the training runs. However, this does not affect the two main trends we can clearly observe: Compute-optimal xLSTM models are larger than their Transformer counterparts and the compute-optimal model size of xLSTMs is robust to variations in context length (optimal model size lines of xLSTM are closer together than Transformer models which are larger apart).
> >
> > Re 3, L399):
> > This statement refers to the two rightmost points in Figure 6 (right). Indeed, it actually not only holds for 16k prefill as we state in line 399, but also for 12k prefill. The main intention behind the statement was to highlight the extreme increase of step-time for Transformers for long prefill lengths, while xLSTM maintains a constant step time, as shown in Figure 6.
> >
> > Re 4):
> > We comment on this in our limitations section and view rigorous long-context evals & benchmarks and comparisons as important future directions. We believe that the scope of carefully analyzing long-context performance across scale warrants a dedicated study.
> >
> > We thank the reviewer for their time and feedback and hope we could clarify all questions and concerns.

---

> > > ### Comment · Reviewer_LrJB · 2025-11-24
> > >
> > > Thanks for the response and clarifications. I will keep my score. Good luck.

---

### Official Review · Reviewer_R1D3 · 2025-10-31

**Soundness:** 3
**Presentation:** 2
**Contribution:** 2
**Rating:** 4
**Confidence:** 4

**Summary:**

This paper systematically compares the scaling of xLSTM and Transformers. Using extensive controlled runs (80M–7B parameters; 2B–2T tokens) and two protocols (IsoFLOP budgeting and parametric fitting), it examines: (i) scaling in compute-optimal and over-training regimes, (ii) how compute-optimal model size depends on context length, and (iii) inference scaling in algorithmic complexity, latency, and throughput. The key finding is that xLSTM consistently scales better than Transformers, and the advantage widens as training or inference contexts grow longer.

**Strengths:**

1.  Provides a large, controlled scaling study comparing xLSTM and Transformers across budgets and context lengths. Combines IsoFLOP budgeting with parametric fitting to bridge compute-optimal and over-training regimes.

2.  Extensive, well-structured experimental sweeps (80M–7B; 2B–2T) with clear loss–compute Pareto analyses.

3.  Coherent problem framing and storyline from scaling laws → compute-optimal sizing → inference scaling. Figures generally align with claims and support the narrative.

4. Offers compute-aware guidance on architecture choice, model size, and context length. Helps quantify when linear-time sequence processing becomes advantageous, informing system design and resource allocation decisions.

**Weaknesses:**

1.  Several key definitions/configs are only provided in the appendix; Moving essential config details into the main text, unifying definitions/notation would significantly improve the presentation of the paper.

2. Results seem to be on a specific data mix, tokenizer, and recipe. Adding cross-dataset, cross-tokenizer, and recipe sensitivity studies would make the paper stronger.

3. Some figures are too crowded. Several plots are visually dense, making it hard to extract key trends.

**Questions:**

1.	It would be good to provide cross-dataset and cross-tokenizer results to test whether the reported margins persist?

2.	How do you normalize FLOPs/MemOps across architectures (kernel fusion, activation checkpointing, flash-attention)?

---

> ### Author Response · Authors · 2025-11-14
>
> We thank reviewer R1D3 for highlighting the extensive scale and coherent framing of our experiments. We are also thankful for their feedback, which helped to improve our paper.
>
> **Weaknesses:**
>
> Re 1):
> We tried to unify the notations and definitions for our scaling law descriptions in Section 2 and 3. Moreover, in Table 4 we provide a unified notation that we use for all FLOP and Memory Op calculations.
> We moved some parts of the hyperparameter and configuration details in the Appendix due to space constraints. In the updated version we will extend the config and hyperparameter details in the main paper, as we get one additional page
> We are not aware of any key definitions/notations/configs that are ambiguous or missing in the main text, but we would happily improve if any are pointed out.
>
> Re 2):
> As we describe in the paper we use a unified data and tokenizer setup for Transformer and xLSTM models as the main focus of this work is a controlled architecture comparison, which is why we control for data and tokenizer. Due to compute constraints, we had to focus on one dataset setup.
> We follow a recipe which is widely adopted: AdamW with linear warmup, cosine decay, and linear cooldown. We do vary training configurations, i.e., we do FLOP controlled experiments (IsoFLOP) and token-per-parameter ratio controlled experiments (Token/Param) which have an impact on the training lengths and learning rate. So to some extent we already vary the training recipe.
> Again our focus was on fair comparison between architecture, this is why we controlled for data + recipe.
>
> Re 3):
> Thank you for the feedback on our figures. We put extensive efforts in plotting and visualizing the large amounts of data we collected and acknowledge that e.g. Figure 5 might appear dense at first sight. We will try to further improve our plots by making them even more clear. Let us know if you have a specific figure in mind.
>
> **Questions:**
>
> Re 1):
> Please see response 2) to the weaknesses above.
>
> Re 2):
> Thank you for the question. We use the same FLOP counts for layers that are identical in Transformer and xLSTM e.g. MLPs and Layernorms. (compare Table 10 & Table 12)
> For Attention we adopted the FLOP calculation from previous work [1].
> See equation 10 with Fsm = 5 & Fcausal = 0.5.
> For the mLSTM we use the chunkwise parallel flop counts with chunk size 64 and Fcausal 1.0 from [2].
> Where applicable we also count non-matmul flops in our mLSTM kernels (e.g. gate activations in the mLSTM), in order to make our FLOP counts as accurate as possible. Thanks again for this question. We will clarify this in the respective sections in the paper.
>
> Kernel fusion and activation checkpointing are training specific optimizations. They mostly impact the wall-clock runtime (e.g. kernel fusion can speed up training, but usually does not change the amount of FLOPs significantly) or reduce memory through recomputation (e.g. activation checkpointing) and are not architecture specific. Therefore we do not account for activation checkpointing in our FLOP calculations. To summarize, we aim to capture the nominal model FLOPs, instead of the effective FLOPs which might vary due to recomputation, which is in line with previous work on scaling laws.
>
> We thank the reviewer again for their helpful feedback, and hope they find their questions addressed.
>
> [1] Dan Busbridge, Amitis Shidani, Floris Weers, Jason Ramapuram, Etai Littwin, and Russ Webb.
> Distillation Scaling Laws. ArXiv, 2502.08606, 2025.
>
> [2] Maximilian Beck, Korbinian Pöppel, Phillip Lippe, and Sepp Hochreiter. Tiled Flash Linear Attention:
> More Efficient Linear RNN and xLSTM Kernels. 2025a. URL https://openreview.net/
> forum?id=b6H64u6TqI.

---

> > ### Comment · Reviewer_R1D3 · 2025-11-27
> >
> > Thank you for the clarification. I hope some of these points will be incorporated into the revised version to further strengthen the work.

---

> > > ### Author Response · Authors · 2025-11-27
> > >
> > > We thank the reviewer for their reply. We will incorporate those points into the final version of the paper and agree that this will strengthen our contribution. Are there any open questions or unaddressed weaknesses remaining that prevent a positive reassessment of our work?

---

### Author Response · Authors · 2025-11-18
**Revision**

Dear Reviewers, we updated the manuscript to incorporate the changes discussed in the first round of rebuttal.
Best, Authors

---

### Meta-Review · Area_Chair_kctX · 2026-01-03

**Summary:**

This paper presents a large empirical study comparing xLSTM and Transformer scaling across model sizes, token budgets, context lengths, and inference metrics, using IsoFLOP and parametric-fit protocols. Reviewers generally agree the experimental sweep is extensive and the study is carefully executed.

The reviewers also noted several concerns that weaken the strength of the paper’s central claims. The main issues are (i) uncertainty about how representative the Transformer baselines are relative to modern strong Transformer variants and the absence of critical high-compute Transformer data points; (ii) limited robustness evidence (single data/tokenizer/recipe) with no cross-dataset/tokenizer sensitivity analysis; and (iii) the paper evaluates only pretraining loss / system metrics without downstream or long-context task validation.

Despite these concerns, the work is overall well-executed and an important contribution to the literature.

**Reviewer Concerns:**

Reviewer R1D3
Addressed / partially addressed by rebuttal: Clarified FLOP counting choices and how identical components are handled across architectures; Committed to unify definitions/notation and move key configuration details toward the main paper where possible; acknowledged figure density and intent to improve clarity.
Still outstanding: results are tied to a single data mix/tokenizer/recipe; no cross-dataset/cross-tokenizer/recipe sensitivity is provided.

Reviewer LrJB
Addressed / clarified by rebuttal: Explained the context-length experiments use packing to a target length and that “nominal tokens” include padding; clarified that the intended comparison is via fitted compute-optimal model size trends rather than directly comparing validation losses across context lengths. Acknowledged the limitation of missing downstream/long-context benchmarks.
Still outstanding: No downstream task evaluation is added;

Reviewer dz4g
Addressed / clarified by rebuttal: Clarified some baseline details; Explained compute constraints preventing an additional very-high-compute Transformer point.
Still outstanding: The core concern remains: the Transformer baseline may be weaker than modern best-practice Transformers; missing a very-high-compute Transformer data point.

Reviewer H1t8
Authors articulated what they view as novel contributions; clarified scope is a deep controlled comparison rather than a broad architecture comparison.
Still outstanding: Limited comparisons to other non-Transformer baselines

**Reviewer Scores:**

R1D3 (score 4): Likely unchanged. Reviewer acknowledged clarifications and suggested incorporating them, but did not indicate an improved score.
LrJB (score 6): Unchanged. Reviewer explicitly stated they would keep their score after rebuttal/clarifications.
dz4g (score 8):  Likely 7. The reviewer indicated their “true” view is between 6 and 8.
H1t8 (score 4): Likely 5.

---

### Decision · Program_Chairs · 2026-01-26

Accept (Poster)